# Biologically Inspired Learning Model for Instructed Vision

**Roy Abel**
Weizmann Institute of Science
`roy.abel@weizmann.ac.il`

**Shimon Ullman**
Weizmann Institute of Science
`shimon.ullman@weizmann.ac.il`

## Abstract

As part of the effort to understand how the brain learns, ongoing research seeks to combine biological knowledge with current artificial intelligence (AI) modeling in an attempt to find an efficient biologically plausible learning scheme. Current models often use a cortical-like combination of bottom-up (BU) and top-down (TD) processing, where the TD part carries feedback signals for learning. However, in the visual cortex, the TD pathway plays a second major role in visual attention, by guiding the visual process toward locations and tasks of interest. A biological model should therefore integrate both learning and visual guidance. We introduce a model that uses a cortical-like combination of BU and TD processing that naturally integrates the two major functions of the TD stream. This integration is achieved through an appropriate connectivity pattern between the BU and TD streams, a novel processing cycle that uses the TD stream twice, and a 'Counter-Hebb' learning mechanism that operates across both streams. We show that the 'Counter-Hebb' mechanism can provide an exact backpropagation synaptic modification. Additionally, our model can effectively guide the visual stream to perform a task of interest, achieving competitive performance on standard multi-task learning benchmarks compared to AI models. The successful combination of learning and visual guidance could provide a new view on combining BU and TD processing in human vision and suggests possible directions for both biologically plausible models and artificial instructed models, such as vision-language models (VLMs).

## 1 Introduction

Understanding how the human brain learns has been a longstanding pursuit in both neuroscience and artificial intelligence (AI). An extensive research area at this intersection is the development of biologically plausible models of cortical learning, particularly in visual processes [Lee et al., 2015, Lillicrap et al., 2016, Scellier and Bengio, 2017, Whittington and Bogacz, 2017, Bozkurt et al., 2024]. While the ultimate goal is to develop a detailed biological model, current research primarily focuses on potential schemes for modifying synaptic weights during learning, including how these modifications are determined and how they propagate through the cortical network (the credit assignment problem) [Whittington and Bogacz, 2019].

Biologically plausible learning models typically incorporate a combination of feedforward (bottom-up) and feedback (top-down) pathways, with the top-down (TD) stream playing a central role in the learning process, similar to the structure of the human cortex [Lillicrap et al., 2020, Song et al., 2021]. However, a key distinction between these models and the cortex is that, while most biological models primarily use the TD stream to propagate feedback signals, in the cortex, the TD stream also plays a crucial role in perception by directing attention [Manita et al., 2015].

Top-down attention is an essential part of human vision and has been extensively studied in human physiology, anatomy, and psychophysics [Treisman and Gelade, 1980, Itti and Koch, 2001, Carrasco,

38th Conference on Neural Information Processing Systems (NeurIPS 2024).

2011]. Studies indicate that the TD stream directs visual processes toward selected locations and tasks, actively shaping the neural activity in the bottom-up (BU) stream and creating task-dependent representations in the cortex [Gilbert and Li, 2013, Harel et al., 2014]. Consequently, current biologically plausible models have been criticized for not involving the TD stream in ongoing visual processes [Lillicrap et al., 2020], and the challenge of incorporating the TD stream into feedforward BU processing remains an open research question [Zagha, 2020, Kreiman and Serre, 2020].

Our work addresses this gap and, to our knowledge, introduces the first biologically plausible learning model where the TD stream not only carries feedback signals but also performs visual guidance. This model tackles two key challenges that must be addressed together: first, *attention guidance*, understanding how the TD stream guides BU neural processing, and second, *learning*, determining synaptic modifications and solving the credit assignment problem in this guided processing context.

The dual role of the TD component in our model, involving both attention guidance and learning, may offer a more accurate depiction of bi-directional cortical processing and learning compared to existing models. For learning, we suggest a 'Counter-Hebb' mechanism, a modification of classical Hebbian learning [Hebb, 2005]. We show that this model can approximate the backpropagation (BP) synaptic modification [Rumelhart et al., 1986], and can provide an exact equivalence to BP under a symmetry assumption. Regarding its biological plausibility, we address the weight symmetry problem between forward and backward paths and use local synaptic updates dependent only on neurons associated with the modified synapse. Moreover, our method offers a possible solution to the long-standing challenge of integrating the TD stream into ongoing visual processing [Zagha, 2020, Kreiman and Serre, 2020, Lillicrap et al., 2020]. In the context of guidance, we demonstrate that the TD stream can be used both for learning and for directing the BU stream to perform tasks of interest by selecting a sparse, task-specific sub-network within the full BU network. We further show that this model achieves competitive results on standard multi-task learning benchmarks.

Beyond brain-related aspects, it is noteworthy that the integration of guidance is becoming a central aspect in recent AI models, particularly in Large Language Models (LLMs) and Vision-Language Models (VLMs). A fundamental characteristic of VLMs, similar to the brain, is their ability to focus on specific tasks of interest [Huang et al., 2023, Liu et al., 2024]. In these models, guidance is achieved through instructions that propagate through a language stream, which interacts with a visual component, allowing the model to dynamically adjust its attention to focus on the particular visual elements relevant to the task. This represents a significant shift from earlier computer vision models, where outputs relied exclusively on visual input and task instructions were implicitly embedded in the model design, with each task handled by a separate model. The parallel between AI advancements and brain modeling may deepen our understanding of the human brain and inspire the development of more advanced, human-like AI systems.

The key contributions of our work include:

- We propose the first biologically motivated learning model for instructed vision.
- We present a unified feedback mechanism that combines error propagation for synaptic learning with Top-Down attention to guide visual processing based on instructions.
- We suggest a Counter-Hebb learning procedure as a possible local synaptic modification that can perform exact backpropagation learning.

The code for reproducing the experiments and creating BU-TD models for guided models is available at `https://github.com/royabel/Top-Down-Networks`.

## 2 Related work

The fields of brain modeling and AI have fruitful interactions going in both directions [Yamins and DiCarlo, 2016, Bowers, 2017, Yildirim et al., 2019]. Particularly, the study of biologically plausible learning models aims to deepen our understanding of the learning mechanisms in the human brain and enhance learning techniques for artificial neural network models. While artificial models primarily employ the backpropagation (BP) algorithm for learning [Rumelhart et al., 1986], a direct implementation of BP in biological models is generally considered biologically implausible [Whittington and Bogacz, 2019, Lillicrap et al., 2020]. Nevertheless, the integration of BP with biological principles, such as Hebb's plasticity rule [Hebb, 2005], has inspired the development of diverse biologically plausible learning approaches.

These learning methods are often compared to learning with BP, aiming to achieve similar performance in a more biologically plausible way. For instance, Equilibrium Propagation methods [Scellier and Bengio, 2017] suggest updating synaptic weights once the model reaches a stable equilibrium state under a given input. Equilibrium Propagation methods have been shown to produce weight updates equivalent to those in BP under specific conditions [Ernoult et al., 2019], and approximate BP in other cases [Millidge et al., 2020]. The Predictive Coding approach suggests that in visual processing, feedback connections carry predictions of neural activities, whereas feedforward streams carry the residual errors between these predictions and the actual neural activities [Rao and Ballard, 1999]. It has been shown that using predictive coding to train a neural network in a supervised learning setting can yield parameter updates that closely approximate those produced by backpropagation [Whittington and Bogacz, 2017, Millidge et al., 2022a]. While subsequent works have further developed these methods [Song et al., 2020, Salvatori et al., 2022], the introduced modifications have faced criticism for reducing their biological plausibility [Rosenbaum, 2022, Golkar et al., 2022].

Among biologically plausible approaches, Feedback Alignment and Target Propagation approaches are the most similar to our method. Like backpropagation, these approaches involve a forward (BU) stream that generates predictions from an input signal, followed by a backward (TD) stream that propagates feedback. While BP propagates gradients backward using the same weights as the forward path, feedback alignment methods propose propagating gradient-like signals through the TD stream with a separate set of weights, removing the symmetric weight structure of BP [Lillicrap et al., 2016, Nøkland, 2016, Song et al., 2021]. Recent findings show that lateral connections between feedforward and feedback streams can self-assemble and adapt, enhancing the biological plausibility of these approaches [Liao et al., 2024]. The target propagation methods suggest propagating backward targets for the forward path instead of gradients [Bengio, 2014, Lee et al., 2015, Meulemans et al., 2020]. Both feedback alignment and target propagation methods can approximate the BP update under specific conditions [Akrout et al., 2019, Ahmad et al., 2020, Ernoult et al., 2022]. Nevertheless, current models lack the extensive BU-TD interactions observed in the brain, which are essential for guiding attention in visual processes [Harel et al., 2014, Manita et al., 2015, Lillicrap et al., 2020].

## 2.1 Guided visual processing

Human cortical processing uses a combination of bottom-up (BU) and top-down (TD) processing streams. In the visual brain, the BU stream proceeds from low-level sensory regions to high-level, more cognitive areas, while in the TD stream processing flows in the opposite direction [Dehaene et al., 2021]. In human vision, the TD stream is involved in TD attention, guiding the visual process and directing it toward tasks of interest [Goddard et al., 2022, Shahdloo et al., 2022]. For example, at the physiological level, it has been shown, in behaving primate studies, that given the same image, but with different tasks, the activation along the BU stream changes, modulated by TD activation, to focus on the instructed task [Gilbert and Li, 2013].

The ability to guide visual processing to extract specific aspects of the image is essential because a single image encompasses a wealth of information regarding objects, their parts and sub-parts, their properties, and inter-relations. Consequently, for complex images, it becomes difficult to extract and represent all the possibly meaningful information through a single visual representation [Huang et al., 2023]. There are two approaches to address this challenge: one is to employ specialized models, each tailored to specific visual tasks, and the other is to develop general-purpose vision models that can selectively focus on relevant visual information

Empirical studies in artificial models have demonstrated that guiding the model's attention to selected locations or tasks offers advantages over non-guided, pure BU models [Tsotsos, 2021, Pang et al., 2021, Ullman et al., 2023]. Furthermore, as opposed to earlier computer vision models, which relied solely on visual inputs without guidance, recent Vision Language Models (VLMs) have integrated instruction guidance mechanisms into their visual processing [Bai et al., 2023, Zhu et al., 2023, Liu et al., 2024, Dai et al., 2024]. The processing of visual information in VLMs integrates a language stream that interacts with a visual stream and guides it to perform selected tasks. As a result, these models have been shown to have high generalization and zero-shot capabilities. The importance of guidance mechanisms in both the human cortex and AI models highlights the need for biologically inspired learning models that incorporate these mechanisms to neuroscience implications and potential advancements of artificial models. While our focus in this paper centers on vision, it is worth noting that our method can be applied to guided processing in other domains as well.

# 3 The bottom-up top-down model

This section introduces the proposed structure of the Bottom-Up (BU) and Top-Down (TD) networks. A BU network with $L$ hidden layers is a function that maps an input vector $x := h_0$ to an output vector $y$, such that for each layer $0 \leq l < L$: the hidden values are defined to be:

$$h_{l+1} := \sigma\left(f_{l+1}(h_0, h_1, ..., h_l)\right) \tag{1}$$

The functions $f_l$ are linear, and the activation function $\sigma$ is an element-wise function that may be non-linear. To predict an output, we use a prediction head $H_{pred}$, which is a small network, typically one to two layers, that maps the last hidden layer $h_L$ to the predicted output: $y = H_{pred}(h_L)$.

For a given BU network, we define a symmetric TD network (denoted with upper bars) as the reverse-architecture network, that maps an input vector $\bar{y}$ to an output vector $\bar{x} := \bar{h}_0$. The TD network is constructed based on the BU architecture as follows: The input (e.g. the prediction error) $\bar{y}$ is mapped to the top-level hidden layer $\bar{h}_L$ of the TD network via the TD prediction head: $\bar{h}_L = \bar{H}_{pred}(\bar{y})$, and then for every $0 \leq l < L$:

$$\bar{h}_l := \bar{\sigma}\left(\bar{f}_{l+1}(\bar{h}_L, \bar{h}_{L-1}, ..., \bar{h}_{l+1})\right) \tag{2}$$

The TD network satisfies two conditions. First, we restrict $\bar{h}_l$ for each $l$ so that $h_l$ and $\bar{h}_l$ will have the same size (the same number of neurons). This allows us to define pairs of corresponding neurons, assigning each BU neuron $h_{l,i}$ in layer $l$ a 'counter neuron' $\bar{h}_{l,i}$ in the TD network. We also use the following notation for simplicity: $\bar{\bar{h}} := h$. Additionally, we restrict $\bar{f}_l$ to have the same connectivity structure as $f_l$, but with the opposite direction: each pair of TD neurons is linked if and only if a link exists between their corresponding BU counter neurons. For example, given a fully connected layer $h_l = f_l(h_{l-1}) = W_l h_{l-1}$, the corresponding TD layer $\bar{f}_l$ is defined to be also a fully connected layer $\bar{h}_{l-1} = \bar{f}_l(\bar{h}_l) = \bar{W}_l \bar{h}_l$ such that the shape of the TD weights matrix $\bar{W}_l$ is equal to the shape of the transposed BU weights matrix $W_l^T$.

## 3.1 Activation functions and biases

The activation functions $\sigma, \bar{\sigma}$, may be any element-wise functions. In this work, we focus on two functions. The first is ReLU which is commonly used for neural networks $ReLU(x) := x \cdot I_{\{x>0\}}$.

The second is the Gated Linear Unit (GaLU) [Fiat et al., 2019], which, in our model, leverages lateral connectivity between the BU and TD streams by gating neurons' activity based on the activation of their counter neurons.

$$GaLU(x) := GaLU(x, \bar{x}) := x \cdot I_{\{\bar{x}>0\}} = \begin{cases} x & \bar{x} > 0 \\ 0 & \bar{x} \leq 0 \end{cases} \tag{3}$$

Where $\bar{x}$ is the counter neuron of $x$ (either a BU or a TD neuron), and $I$ is an indicator function.

GaLU introduces bidirectional lateral connectivity between the BU and TD networks by temporarily turning off neurons based on the values of their counter neurons. As a result, each network can effectively guide its counterpart to operate on a specific partial sub-network.

In this paper, we omit bias terms to simplify the model. Nevertheless, biases can be implicitly expressed using the above notations by having additional neurons and weights, as commonly practiced [Lee et al., 2015, Ahmad et al., 2020]. In addition, we allow two modes of biases. The first is the standard bias mechanism, in which biases contribute to the output. The second mode is 'bias-blocking' [Akrout et al., 2019] in which all bias terms are zeroed.

# 4 Counter-Hebbian learning

In this section, we formulate the Counter-Hebb learning. The Counter-Hebb (CH) rule updates each synapse based on the activities of its pre-synaptic neuron and its post-synaptic counter neuron. Consider a given weights matrix $W$ such that $b = Wa$. The $(i,j)$-th entry in that matrix, $W_{ij}^{(t)}$, represents the strength of the synapse connecting the pre-synaptic neuron $a_j$ to the post-synaptic neuron $b_i$ at time $t$. Then the update rule is:

$$\Delta W_{ij}^{(t+1)} := W_{ij}^{(t+1)} - W_{ij}^{(t)} = \eta \cdot a_j \cdot \bar{b}_i \tag{4}$$

where $\eta$ is the learning rate, and $\bar{b}_i$ is the counter neuron of $b_i$. This rule applies to all weights including both up-streams and down-streams, updating both $W$ and $\bar{W}$ identically, see Figure 1.

There is a close connection between this rule and the classic Hebb rule. In both cases, the brain strengthens synapses (weights) between neurons that are co-activated, and the modifications of each synapse are determined entirely by the activation values of neurons in the network associated directly with the changing synapse. However, the difference lies in the neurons connected to that synapse. Classic Hebbian proposes that the forward-firing of a post-synaptic neuron also propagates backward to the synapse. Thereby, synapse strength increases when a pre-synaptic neuron's firing is often followed by the firing of the post-synaptic neuron within a defined time interval [Magee and Johnston, 1997, Hebb, 2005]. In contrast, the Counter-Hebb modification does not depend on the cell's firing propagating back to its dendrites but suggests a contribution from the counter post-synaptic neuron via a lateral connection. Similar to Hebb's rule, the resulting synaptic modification also depends on the coincidence of two firing neurons, but the post-synaptic cell is replaced by its counterpart. See Figure 1 for a visual illustration of the Counter-Hebb update compared with the classic Hebb.

Therefore, the CH rule modifies the classical Hebb rule by integrating feedback streams that can carry error information into the learning process. The feasibility of synaptic plasticity that depends on the coincidence of two signals from feedforward and feedback sources is supported by empirical findings, such as those from CA1 hippocampal cells [Markov et al., 2014, Cornford et al., 2019].

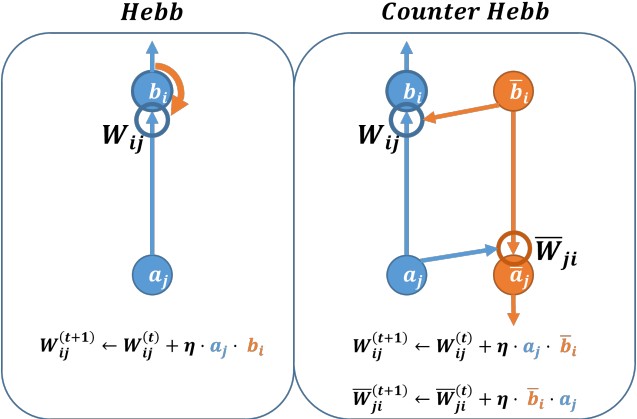

Figure 1: The Counter-Hebb update rule in comparison with the classical Hebb rule. The classical Hebb rule (on the left), with a focus on a single upstream synapse $W_{ij}$ (outlined by a circle), connecting a pre-synaptic neuron $a_j$ with a post-synaptic neuron $b_i$. The synapse $W_{ij}$ is updated based on the activity of both associated neurons $a_j$ and $b_i$. While neuron $a_j$ is directly associated with the synapse $W_{ij}$, neuron $b_i$ is assumed to transmit its information through propagation down the dendritic tree to synapse $W_{ij}$ (orange arrow). In contrast, the Counter-Hebb update rule (on the right), relies on a contribution from the counterpart downstream (marked in orange), mediated via lateral connections. Compared with the Hebb rule, the signal from $a_j$ is combined with the signal from neuron $\bar{b}_i$ rather than neuron $b_i$. Notably, the resulting Counter-Hebb rule naturally applies an identical update to both $W_{ij}$ and its counter synapse $\bar{W}_{ji}$.

## 4.1 The Counter-Hebb learning algorithm and backpropagation

This section presents the full Counter-Hebbian (CH) learning algorithm. The learning algorithm is described in Algorithm 1. Similar to the backpropagation algorithm, the CH algorithm involves a single forward pass performed by the BU network to compute predictions from an input signal. Subsequently, a single backward pass is conducted using the TD network to propagate error information, and the weights are updated according to the CH update rule.

A special case occurs when the BU and TD networks have symmetric weights, (identical values). While symmetric BU and TD weights might, at first, seem unrealistic in the brain, symmetry is actually implicitly encouraged by the CH update. The CH update naturally applies an identical update to both the BU and the TD weights, see Figure 1. Therefore, as training progresses, assuming close-to-zero initial weights (a common practice), the BU and TD weights gradually become more

---

**Algorithm 1** Counter-Hebb Learning

---
1: **Input:** data $x$, ground truth label $\tilde{y}$
2: **Forward**: $y = BU(x)$
3: **Compute error**: $e = error(y, \tilde{y})$
4: **Backward**: $\bar{x} = TD(e)$
5: **Counter-Hebb Update**: update $W$ and $\bar{W}$

---

symmetric, as the value of the weights will be dominant by the values of the updates. Additionally, at any point during the training, if the BU and TD weights are symmetric, they will maintain this symmetry during the entire learning.

Given symmetric BU and TD weights, under the following standard conditions: 1) The BU network uses ReLU non-linearity 2) The error function computes the negative gradients of a loss function $L$ with respect to the BU output, for example, $error(y, \tilde{y}) = \tilde{y} - y$ for Mean Squared Error loss 3) The TD network uses GaLU non-linearity and bias-blocking mode (see Section 3.1). Then the TD backward step in Algorithm 1 is mathematically equivalent to the backward computation of the BP algorithm [Rumelhart et al., 1986]. As a result, in this configuration, Counter-Hebb learning effectively replicates the exact BP update, performing similarly to BP and preserving its mathematical properties. Moreover, relaxing the symmetry constraint under the above conditions results in a learning algorithm that approximates BP in the non-symmetric case. For a detailed explanation of this equivalence and approximation see Appendix A.1. Like some previous models, the CH has the desired property, as a biological model, of locality: the synaptic modifications are determined entirely by the activation values of neurons directly associated with the synapse.

## 5 Instruction-based learning

In the previous section, we described how the TD network can be used for learning a pure BU model. In this section, we describe how the model performs visual guidance. The TD network in our model can guide the BU network to perform multiple tasks by selecting a sub-network for each learned task (where sub-networks can overlap). In this setting, the objective is to predict an output $y$ given an input $x$ and a task $t$. To accommodate this, the model has one additional head, resulting in two heads: a prediction head, and an instruction head. Each head consists of two parts: one for the BU network and the other for the TD network, preserving the symmetrical structure and lateral connectivity of the BU-TD core, see Appendix A.2 for more details on the heads.

The prediction head $H_{pred}$, of one linear layer, is responsible for generating predictions and providing feedback, as discussed in section 3. The instruction head, $H_{instruct}$, employs a 2-layer MLP for specifying the selected task, projecting instructional information to the visual space (and vice versa), similarly to instruction processing in VLMs. More specifically, the instruction head takes a task representation $t$ as input and maps it to the top-level TD layer $\bar{h}_L$. We use one-hot encoding for representing the tasks, however, more complex embedding could also be explored, such as a projection from an LLM. Note that in our experiments, we allow only one head to participate in each pass of the model (either the prediction or instruction head), refer to Fig 2 for an illustration of how the two heads are utilized in learning instruction-based models.

The instruction-based learning algorithm is shown in Algorithm 2. This algorithm consists of two passes for prediction (a TD followed by BU) followed by an additional TD pass for the learning, thereby extending Algorithm 1 by adding an instruction-processing step that selects a task-specific sub-network. Given a task $t$, the TD instruction head is used to propagate the task representation along the TD network. Since each task activates different patterns, the activated neurons (i.e., with activation value larger than 0) define a task-dependent sub-network. By running the BU network with GaLU activation, the BU computation is gated to propagate the input $x$ along the corresponding BU sub-network. In this manner, the resulting algorithm learns for each task a different predictor which is conditioned on the task, resembling a modular architecture where different modules are dedicated to each task. See Fig 2 for a visualization of this guided process.

Extending upon the results of section 4.1, with the constraint of symmetric BU and TD weights, both BP and CH learning produce identical updates within this guided learning framework, despite using the TD stream twice, see Appendix A.1 for more details. This equivalence provides mathematical

---
**Algorithm 2** Instruction-Based Learning
---
1: **Input:** data $x$, task $t$, ground truth label $\tilde{y}$
2: **Top-Down**: $\bar{x} = TD(t; \; \bar{H}_{instruct}; \; \bar{\sigma} = ReLU)$
3: **Bottom-Up**: $y = BU(x; \; H_{pred}; \; \sigma = GaLU \circ ReLU)$
4: **Compute error**: $e = error(y, \tilde{y})$
5: **Backward**: $\bar{x} = TD(e; \; \bar{H}_{pred}; \; \bar{\sigma} = GaLU)$
6: **Counter-Hebb Update**: update $W$ and $\bar{W}$
---

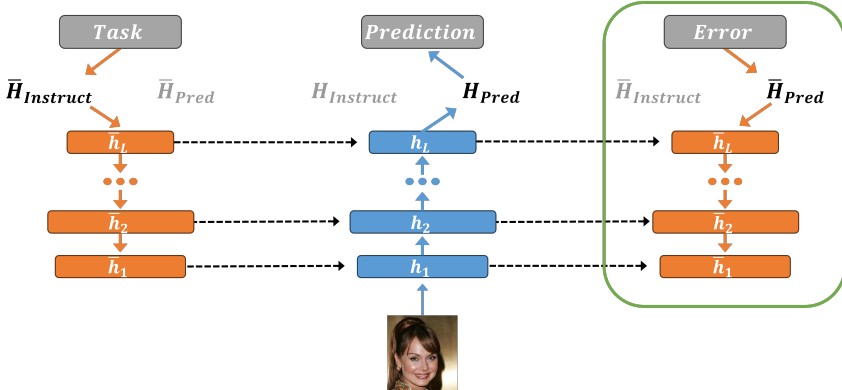

Figure 2: The instruction-based learning algorithm. The three columns represent three passes of our model (left to right): $TD \rightarrow BU \rightarrow TD$, where the first two passes provide a prediction output given an image and a task, and the last TD pass (in green frame) is used for learning. In *inference*, The BU visual process is guided by the TD network according to the given task. More specifically, The TD network propagates instruction signals downward followed by a guided BU process of the input image to compute predictions. By applying ReLU non-linearity, the input task selectively activates a subset of neurons (i.e. non-zero values), composing a sub-network within the full network. The BU network then processes an input image using a composition of ReLU and GaLU. The GaLU function (dashed arrows) gates the BU computation to operate only on the selected sub-network that was activated by the task. For *learning*, the same TD network is then reused to propagate prediction error signals with GaLU exclusively (no ReLU). Finally, the 'Counter-Hebb' learning rule adjusts both networks' weights based on the activation values of their neurons. Therefore, in contrast with standard models, the entire computation, including the learning, is carried out by neurons in the network, and no additional computation is used for learning (e.g. backpropagation)

guarantees to learning guided vision using a single TD network for both guidance and learning. Furthermore, symmetric weights have computational advantages. It enables extending standard BU architectures to instruction-based models without any additional parameters. For a given BU network, a complementary symmetric TD network can be constructed, sharing the same BU parameters. This TD network can guide the BU process of the original network to perform a given instruction.

## 6 Empirical results

In this section, we evaluate our BU-TD model, learned via Counter-Hebbian learning, in two settings: 1) unguided visual processing, to show that CH learning is capable of learning vision models 2) guided visual processing, to evaluate the ability of our model to guide the visual process according to instructions. Our goal is not to improve upon state-of-the-art models, but to show that the model, with a single top-down pathway for both error and instruction propagation, is comparable with current AI models, and capable of performing well two different functions: learning and directing attention.

### 6.1 Unguided visual processing

In the unguided experiments, we evaluate the performance of the Counter-Hebb learning on standard image classification benchmarks: MNIST [LeCun et al., 1998], Fashion-MNIST [Xiao et al., 2017],

Table 1: Unguided learning results: mean and standard deviation of the test accuracy (in percentages) across 10 runs. The proposed CH learning algorithm is compared with BP and other biological state-of-the-art methods. The baseline results were taken from Bozkurt et al. [2024].

| Method | MNIST | Fashion MNIST | CIFAR10 |
|---|---|---|---|
| CIM [2024] | $97.71 \pm 0.1$ | $88.14 \pm 0.3$ | $51.86 \pm 0.3$ |
| EP [2017] | $97.61 \pm 0.1$ | $88.06 \pm 0.7$ | $49.28 \pm 0.5$ |
| CSM [2021] | $98.08 \pm 0.1$ | $88.73 \pm 0.2$ | $40.79$ |
| PC [2017] | $98.17 \pm 0.2$ | $89.31 \pm 0.4$ | - |
| PC-Nudge [2022b] | $97.71 \pm 0.1$ | $88.49 \pm 0.3$ | $48.58 \pm 0.7$ |
| FA [2016] | $97.95 \pm 0.08$ | $88.38 \pm 0.9$ | $52.37 \pm 0.4$ |
| BP | $98.27 \pm 0.03$ | $89.41 \pm 0.2$ | $53.96 \pm 0.3$ |
| BP (ours) | $98.33 \pm 0.04$ | $89.94 \pm 0.2$ | $55.47 \pm 0.3$ |
| CH Sym Init | $98.34 \pm 0.06$ | $89.99 \pm 0.2$ | $55.54 \pm 0.3$ |
| CH Asym Init | $98.17 \pm 0.06$ | $89.27 \pm 0.1$ | $54.28 \pm 0.2$ |

and CIFAR10 [Krizhevsky et al., 2009]. We followed the same experiments as Bozkurt et al. [2024] and used two-layer fully connected networks, with a hidden layer of size 500 for both MNIST and Fashion-MNIST datasets and size 1,000 for CIFAR10. Further details including the full set of hyperparameters can be found in Appendix A.4.2. We compare CH learning using the Cross-Entropy loss with backpropagation and other biological learning methods.

We examine two settings of CH learning, one where the BU and TD weights are initialized with symmetrical values, denoted as 'Sym Init', and the other where the weights are initialized differently, referred to as 'Asym Init'. The results, shown in Table 1, empirically validate that CH learning is equivalent to BP in the symmetric case, and approximates BP in the asymmetric case. Moreover, CH learning achieves comparable or superior performance compared with other biological methods. We further show the robustness of CH on other architectures and settings, such as convolutional networks, loss functions, and regularization. The results and additional information regarding these experiments can be found in Appendix A.4.2.

## 6.2 Guided visual processing

In the guided experiments, we evaluate our model on two common multi-task learning (MTL) benchmarks. Since current biological methods are not capable of guided processing, we compare CH with non-biological state-of-the-art optimization methods as reported by Kurin et al. [2022], replicating their setup and using their reported results.

**The Multi-MNIST** dataset contains images of two overlaid digits, where the task indicates whether to classify the left or the right digit. Similar to the baselines, our BU network employs a simple architecture composed of two convolutional layers followed by a single fully-connected layer, with ReLU non-linearity, along with an additional fully-connected linear layer as the decoder (prediction head). To adapt this architecture to the BU-TD structure, we replace all max-pool layers with strided convolution layers, that perform a similar function as proposed by [Ayachi et al., 2020]. Since the BU-TD model uses only sparse sub-networks within the full network, we increased the number of channels in each convolution layer, however, the actual network size is effectively smaller compared with the baselines, see Appendix A.6 for an analysis of the actual size of the sub-networks.

**The CelebA** dataset is a more challenging large-scale benchmark, comprising head shots of celebrities, along with the indication of the presence or absence of 40 different attributes. Each task is a binary classification problem for an attribute. As done in previous work [Kurin et al., 2022], we employ a ResNet-18 [He et al., 2016] architecture (without the final layer) with batch normalization layers [Ioffe and Szegedy, 2015], and a linear decoder. Additionally, we remove the last average pooling layer to support the symmetric BU-TD structure.

Unlike the baselines, we do not use any learning 'tricks' such as dropout layers, regularization, or early stopping. Additionally, while the baseline models require a separate decoder (prediction head) for each task, our BU-TD model can use a single shared decoder for all tasks. Further details,

Table 2: Guided processing results: mean and 95% confidence interval of the avg. task test accuracy (in percentages) across 10 runs for *Multi-MNIST* and 5 runs for *CelebA*. The proposed CH learning algorithm is compared with non-biological state-of-the-art methods, as reported in Kurin et al..

| Method | Multi-MNIST | CelebA |
|---|---|---|
| BP (Unit. Scal. [2022]) | $94.76 \pm 0.44$ | $90.90 \pm 0.08$ |
| IMTL [2021] | $94.87 \pm 0.25$ | $90.93 \pm 0.08$ |
| MGDA [2018] | $94.78 \pm 0.20$ | $90.22 \pm 0.10$ |
| GradDrop [2020] | $93.47 \pm 1.30$ | $90.98 \pm 0.03$ |
| PCGrad [2020] | $94.79 \pm 0.36$ | $90.93 \pm 0.11$ |
| RLW Diri. [2021] | $94.30 \pm 0.30$ | $90.99 \pm 0.08$ |
| RLW Norm. [2021] | $93.99 \pm 0.89$ | $90.95 \pm 0.10$ |
| CH Asym Init | $88.92 \pm 2.15$ | $79.25 \pm 1.63$ |
| CH Sym Init | $94.20 \pm 0.30$ | $89.69 \pm 0.12$ |

including exact architectures, hyper-parameters, and additional experiments with a single decoder, and varied network sizes, are provided in Appendix A.4.

The results, presented in Table 2, show that the proposed model successfully incorporates the two different TD functions, directing attention, and learning. The BU-TD model can achieve competitive performance compared with leading non-biological state-of-the-art methods. The proposed method may offer additional useful computational properties, such as compactness, see Appendix A.6.

### 6.3 Weight symmetry

Weight symmetry poses a significant challenge in biological learning models, as copying the same weights across different locations is unrealistic in the brain, referred to as the 'weight transport' problem Grossberg [1987]. Therefore, unlike backpropagation, biological models use different weights for feedforward and feedback streams. In this section, we further explore the effect of deviation from weight symmetry on the model performance, focusing on both symmetry in the initialization of the weights, and symmetry in the subsequent updates. We use below the terms 'symmetric model' for models with symmetric weight initialization, 'asymmetric models' for weights initialized far from symmetry, and 'weak symmetric' for models initialized symmetrically (or nearly symmetric), but are subject to noisy, asymmetric updates. See Appendix A.5 for more details and additional experiments.

The results, presented in Table 3 and Appendix A.5, demonstrate that initial weight symmetry is more critical for approximating backpropagation performance than gradually converging to symmetry later in the learning process. The experiments show that performance degradation in the asymmetric case becomes more pronounced as task complexity and model size increase, even with weight decay applied to ensure convergence to symmetry. The results also show that our model demonstrates better scalability with asymmetric weights compared with some alternative methods, such as feedback alignment. In contrast, in the weak symmetric case, where weights do not maintain symmetry due to noise in updates, performance consistently remains nearly identical to standard back-propagation across all experiments. These results indicate that exact weight symmetry is not mandatory for backpropagation approximation. We hypothesize that Counter-Hebb learning, like backpropagation, depends on proper weight initialization to achieve optimal results, in addition to weight symmetry. By the time that symmetry of the weights is obtained, the weights may drift from their optimal initialization. This case will be similar to starting the standard backpropagation from non-optimal initialization conditions, leading in both cases to suboptimal performance. For more details regarding the effects of asymmetry see Appendix A.5.1.

## 7 Limitations

There are two directions that should be improved in the current model, one regarding performance and the second concerning biological aspects. In the asymmetric case, similar to other biologically inspired methods [Ernoult et al., 2022], we observe increasing performance gaps as task and model complexity

Table 3: Comparing different weight asymmetry settings: mean accuracy and 95% confidence interval (in percentages) across 5 runs evaluated on CIFAR10 using ResNet18.

| Method | Train Accuracy | Test Accuracy |
|---|---|---|
| CH Symmetric (BP) | $100.00 \pm 0.00$ | $72.06 \pm 0.69$ |
| CH Asymmetric | $99.70 \pm 0.15$ | $61.58 \pm 0.79$ |
| CH Asymmetric + WD | $99.77 \pm 0.18$ | $62.05 \pm 1.03$ |
| CH Weak Symmetric | $100.00 \pm 0.00$ | $71.98 \pm 0.17$ |
| Feedback Alignment | $22.79 \pm 1.90$ | $23.11 \pm 2.01$ |

increase. However, the learning parameters, including the weight initialization scheme, have been carefully optimized for backpropagation (symmetric weights) over the years. Therefore, examining the effects of tuning parameters in the asymmetric case could improve performance. On the biological side, the plausibility of the model should be further explored. The main issue we identify (in ours and other models) is that propagating error signals along the TD stream requires the representation of both positive and negative values in neuronal activity [Lillicrap et al., 2020]. Following initial work, we suggest that this could be obtained by 'on' and 'off' channels [Ringach, 2004], where negative values in the 'on' channel are represented by positive values in the complementary 'off' channel.

## 8 Discussion

In this paper, we proposed the first biologically-motivated learning model for instructed visual processing. Similar to the visual cortex, it uses a bottom-up (BU) top-down (TD) structure, which, unlike previous learning models, uses the TD stream in ongoing visual processing by directing attention, e.g. to tasks and locations of interest.

Modeling guided visual learning is challenging since the prediction of the model depends on both the BU processing of the image and the task selected by a top-down instruction. The error signal needs, therefore, to propagate through both the BU and TD pathways, and at the same time the network is required to preserve the neural activations from the prediction phase since they determine the required changes in synaptic weights (Fig. 1). These requirements place significant constraints on the structure of the model network, however, our model meets the requirements and, as supported by mathematical foundations and empirical experiments, succeeds in learning guided vision for multiple tasks. Since the cortex performs similar guided vision, the proposed model may suggest a sketch model for the combination of the BU and TD streams in the visual cortex. Furthermore, the model shares a similar general structure with VLMs, in the sense of using two parallel streams, a visual one together with a more cognitive one. Since the human brain excels at combining visual and cognitive information in visual perception, the combination of instructed VLMs with principles from human BU-TD processing can offer a promising direction for future studies.

The model also suggests a Counter-Hebbian learning process in addition to the classical Hebb rule, where synapses are modified by combining a pre-synaptic signal with a signal coming from the appropriate counter stream. The existence of CH learning may be tested biologically by the controlled activation of selected layers. For example, cortical layer 3B receives feedforward connections from layer 4 while feedback connections arrive to layer 2/3A [Markov et al., 2014]. The counter-Hebbian model predicts that it may be possible to modify the forward synapses from layer 4 to layer 3B by simultaneous activation of the two inputs.

Counter-Hebb learning provides new directions for addressing the weight symmetry problem in biological learning models. Our findings indicate that exact weight symmetry is not crucial for achieving performance comparable to backpropagation and performance is more significantly influenced by the choice of weight initialization than by the precise symmetry in the weights. This suggests that discussions around biological feasibility in learning models should focus on obtaining nearly symmetric weights at initialization, rather than starting with asymmetric weights and relying on convergence to symmetry.

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

# A  Appendix

## A.1  Equivalent to the backpropagation algorithm

In this section, we give a detailed explanation for the equivalence between the proposed Counter-Hebbian (CH) learning and Back-Propagation (BP), discussed in section 4.1. BP is a key component of current learning algorithms for artificial neural networks, and modern deep learning models are typically optimized using end-to-end BP and a global loss function [LeCun et al., 2015]. BP is an efficient algorithm to compute gradients, designed especially for deep neural networks [Rumelhart et al., 1986]. The algorithm uses the chain rule to back-propagate error signals through the network, thus computing the gradients of a loss function $L$ with respect to all parameters through a single backward pass.

### A.1.1  Symmetric weights

Given a feedforward BU network architecture, as described in section 3, Then, the BP backward pass propagates error signals $\delta$ through the network from the output layer according to:

$$\delta_{l-1} := \frac{\partial f_l}{\partial h_{l-1}} \delta_l = \sigma'(h_{l-1}) W_l^T \delta_l \tag{5}$$

Where the initial $\delta$ values, that correspond to the output layer, are the derivative of the loss with respect to the output layer: $\delta_L = \frac{\partial \text{Loss}}{\partial h_L}$. This construction of $\delta$ enables an easy way to compute the gradients with respect to each parameter:

$$\nabla \text{Loss}(W_l) = \frac{\partial L}{\partial W_l} = \delta_l h_{l-1}^T \tag{6}$$

Given the following conditions:

- Symmetric BU and TD weights, $W = \bar{W}^T$
- The BU network uses ReLU non-linearity, $\sigma = ReLU$
- The error function computes the negative gradients of a loss function $L$ with respect to the BU output, $e = error(y, \tilde{y}) = -\frac{\partial L(y, \tilde{y})}{\partial y}$
- The TD network uses GaLU non-linearity, $\bar{\sigma} = GaLU$, and bias-blocking mode (see Section 3.1)

The TD process done in the backward step in Algorithm 1 makes the exact same computation as BP at each layer:

$$\bar{h}_{l-1} := \text{GaLU}(\bar{W}_l \bar{h}_l) = \sigma'(h_{l-1}) W_l^T \bar{h}_l \tag{7}$$

This similarity is since the BU and TD weights are symmetric, i.e. $W_l^T = \bar{W}_l$, and the GaLU function effectively applies a product of $x$ with an indicator function which is exactly the gradient of the ReLU function, thus the GaLU operation is equivalent to multiplication with the derivatives of the BU ReLU function.

Therefore, since the input to the TD network is the negative derivative of the loss function with respect to the output, the TD neurons have the same values as the BP signals, up to a different sign:

$$\bar{h}_l = -\delta_l \tag{8}$$

As a result, the update derived from our CH learning is equivalent to BP in this symmetric case and also performs a Gradient Descent (GD) update:

$$\Delta W_l = \eta \bar{h}_l h_{l-1}^T = -\eta \delta_l h_{l-1}^T = -\eta \nabla \text{Loss}(W_l) \tag{9}$$

Moreover, when exploring the non-symmetric case, which has the same conditions as above but the symmetric constraint, we get that CH learning approximates the BP update as the learning progresses.

### A.1.2 Asymmetric weights

In the asymmetric case, the BU and TD weights are initialized with different values. Consider a BU weight matrix $W$ and its counter TD weights $\bar{W}$, where both weights were initialized i.i.d from a uniform distribution $U[-a, a]$, where $a$ is a small positive scalar (a common practice). At each time step $t$ during the learning, the Counter-Hebb update applies a symmetrical update (up to the transposed dimensions of the matrices):

$$\Delta W^{(t)} = \Delta \bar{W}^{(t)} \tag{10}$$

Consequently, in each time step, the difference between the two weight matrices remains constant, and is determined by the initialization of the weight, thus is bounded:

$$\left| W_{ij}^{(t)} - \bar{W}_{ji}^{(t)} \right| = \left| \left( W_{ij}^{(0)} + \sum_{k=1}^{t} \Delta W_{ij}^{(k)} \right) - \left( \bar{W}_{ji}^{(0)} + \sum_{k=1}^{t} \Delta \bar{W}_{ji}^{(k)} \right) \right| =$$
$$= \left| W_{ij}^{(0)} - \bar{W}_{ji}^{(0)} \right| <= 2a$$

Notably, the weight initialization scheme is controlled, therefore the value $a$ can be controlled. Furthermore, a common belief is that high-magnitude weights of a trained network, are the weights that are important for the learned task. Pruning techniques have shown that those weights alone are sufficient for achieving results as good a full model consisting of all the weights [Frankle and Carbin, 2018]. Hence, focusing on a specific important weight (that has a high magnitude) $\left| W_{ij}^{(t)} \right| \gg 0$, then

$$\left| \frac{W_{ij}^{(t)} - \bar{W}_{ji}^{(t)}}{W_{ij}^{(t)}} \right| <= \left| \frac{2a}{W_{ij}^{(t)}} \right| \approx 0$$

Therefore, as the training progresses, assuming close to zero weight initialization, the difference between the BU and TD weights will be negligible for the dominant BU weights that are important for the task. Consequently, as the training proceeds, the Counter-Hebb learning gradually pushes the BU and TD weights towards symmetry, and the CH update rule approximates the BP update.

Moreover, we can make the weights converge to exact symmetry by adding a weight decay mechanism. Denoted the original update at time $t$ by $A(t)$, the new updates at time $t$ will be $\Delta W_l^{(t)} = A(t) - \lambda W^{(t)}$ and $\Delta \bar{W}_l^{(t)T} = A(t) - \lambda \bar{W}_l^{(t)T}$. Thus,

$$\left| W^{(t)} - \bar{W}^{(t)T} \right| = \left| (1 - \lambda)^t W^{(0)} - (1 - \lambda)^t \bar{W}^{(0)T} \right| \xrightarrow{t \to \infty} 0$$

Therefore, similarly to the results shown by Akrout et al. [2019], initializing the BU and TD weights with different values will converge to symmetric BU and TD weights, in which the CH learning is equivalent to backpropagation. Hence, in that non-symmetric case, the Counter-Hebb learning algorithm approximates the BP and approaches the exact BP.

### A.1.3 Guided visual processing

Extending upon the above results to the guided learning framework, under the same constraints of the symmetric case, both BP and CH learning yield identical updates.

The guided learning algorithm consists of two passes for prediction, a TD pass followed by a BU pass. Hence, updating this model via BP requires computing the gradients of the loss function with respect to both the TD and BU computations.

Notably, the first TD computation in the prediction phase, is connected to the final prediction only through the gating functions on the computation graph, see Figure 2. Moreover, the gradients of this function with respect to the gate $\bar{x}$ are always zero. Therefore, this TD computation does not contribute any gradients to the prediction process:

$$\nabla \text{Loss}(\bar{W}_l) = \frac{\partial L}{\partial \bar{W}_l} = 0 \tag{11}$$

Focusing on the BU computation, given the constraint of symmetric BU and TD weights, the results from the non-guided scenario indicate that the last TD pass in our algorithm, used for error propagation, computes the exact backpropagation signals relative to the BU computation.

Consequently, given the constraint of symmetric weights, for example, obtained by sharing the same weights across the two streams, the BP algorithm actually updates both the BU and TD weights according only to the gradients of the BU pass. Hence the backpropagation update is identical to the Counter-Hebb learning update, and we got an equivalence in the guided processing framework.

## A.2 The model heads

In this section, we describe the structure of the heads in our proposed BU-TD model. The BU-TD core network consists of two symmetric neural networks that are connected via lateral connections, as described in 3. This core network is extended by two heads: a prediction head, and an instruction head, each employing an additional small BU-TD neural network. The instruction head employs a 2-layer Multi-Layer Perceptron (MLP), while the prediction head utilizes a single linear layer. Similar to the core, the heads consist of two connected parts: one for the BU network and the other for the TD network, thus preserving the symmetrical structure and lateral connectivity of the BU-TD model.

This results in two pairs of symmetric heads. The first pair is for the predictions: $H_{pred}$ in the BU stream, and its symmetric counterpart in the TD stream $\bar{H}_{pred}$. The second pair is for the instructions: $H_{instruct}$ in the BU stream, and its symmetric counterpart in the TD stream $\bar{H}_{instruct}$. The prediction head is responsible for model predictions. In the BU stream, it generates predictions based on input data, while in the TD stream, it delivers prediction error information. On the other hand, the instruction head bridges the instructional space with visual concepts. The TD stream maps task representations into the model's hidden space, while the BU stream maps the visual space into the instructional space.

Only one head can participate in each pass of the network (either BU pass or TD pass), where heads can be alternated, with a different head chosen in each pass. For instance in the Counter-Hebb guided learning algorithm, the first TD pass uses the instruction head, while the following two passes (BU followed by a TD) use the prediction head.

## A.3 Datasets

Similar to other biologically motivated learning methods, we compare CH learning with BP on standard image classification benchmarks: MNIST [LeCun et al., 1998], Fashion-MNIST [Xiao et al., 2017], and CIFAR10 [Krizhevsky et al., 2009]. In addition, we use two common multi-task learning (MTL) benchmarks, the *Multi-MNIST* [Sabour et al., 2017] dataset, and the *CelebA* [Liu et al., 2015] dataset, to evaluate the ability of our model to guide the visual process according to instruction signals.

**Multi-MNIST**, introduced by [Sabour et al., 2017] and modified by Sener and Koltun [2018], is a simple two-task supervised learning benchmark dataset constructed by uniformly sampling two overlayed MNIST [LeCun et al., 1998] digits. One digit is placed in the top-left corner, while the other is in the bottom-right corner. Each of the two overlaid images corresponds to a 10-class classification task. We generated the dataset using the code provided by Kurin et al. [2022], which samples the training set from the first 50,000 MNIST training images, and the test set from the original MNIST test set. We omitted the validation set, and the hyper-parameters were tuned based solely on the training set.

The **CelebA** dataset [Liu et al., 2015] (with standard training, and test splits) comprises more than 200,000 face images of celebrities along with annotations for 40 attributes, such as the presence of eyeglasses, gender, smiling, and more. Within the context of Multi-Task Learning research, it is frequently approached as a 40-task classification challenge, where each task involves binary classification for one of the attributes.

Table 4: Unguided learning results: mean and standard deviation of the test accuracy (in percentages) across 10 runs. The proposed CH learning algorithm is compared with BP and other biological state-of-the-art methods. The baseline results were taken from Bozkurt et al. [2024].

| Method | MNIST | Fashion MNIST | CIFAR10 |
|---|---|---|---|
| CIM [2024] | $97.71 \pm 0.1$ | $88.14 \pm 0.3$ | $51.86 \pm 0.3$ |
| EP [2017] | $97.61 \pm 0.1$ | $88.06 \pm 0.7$ | $49.28 \pm 0.5$ |
| CSM [2021] | $98.08 \pm 0.1$ | $88.73 \pm 0.2$ | $40.79$ |
| PC [2017] | $98.17 \pm 0.2$ | $89.31 \pm 0.4$ | - |
| PC-Nudge [2022b] | $97.71 \pm 0.1$ | $88.49 \pm 0.3$ | $48.58 \pm 0.7$ |
| FA [2016] (Cross-Entropy loss) | $97.95 \pm 0.08$ | $88.38 \pm 0.9$ | $52.37 \pm 0.4$ |
| FA [2016] (MSE loss) | $97.99 \pm 0.03$ | $88.72 \pm 0.5$ | $50.75 \pm 0.4$ |
| BP (Cross-Entropy loss) | $98.27 \pm 0.03$ | $89.41 \pm 0.2$ | $53.96 \pm 0.3$ |
| BP (MSE loss) | $97.58 \pm 0.01$ | $88.39 \pm 0.1$ | $52.75 \pm 0.1$ |
| BP (ours) (Cross-Entropy loss) | $98.33 \pm 0.04$ | $89.94 \pm 0.2$ | $55.47 \pm 0.3$ |
| CH Sym Init (Cross-Entropy loss) | $98.34 \pm 0.06$ | $89.99 \pm 0.2$ | $55.54 \pm 0.3$ |
| CH Asym Init (Cross-Entropy loss) | $98.17 \pm 0.06$ | $89.27 \pm 0.1$ | $54.28 \pm 0.2$ |
| BP (ours) (MSE loss) | $98.36 \pm 0.08$ | $90.16 \pm 0.2$ | $54.50 \pm 0.4$ |
| CH Sym Init (MSE loss) | $98.37 \pm 0.07$ | $90.13 \pm 0.2$ | $54.56 \pm 0.3$ |
| CH Asym Init (MSE loss) | $98.21 \pm 0.06$ | $89.54 \pm 0.2$ | $53.09 \pm 0.3$ |

## A.4 Experimental settings and results

### A.4.1 Computational resources

All the experiments were conducted using either NVIDIA RTX 6000 GPU or NVIDIA RTX 8000 GPU. For all experiments but CelebA, a single NVIDIA RTX 6000 GPU was used, with the experiments utilizing only a fraction of its capacity. In the case of the CelebA dataset, either a single NVIDIA RTX 8000 GPU or two NVIDIA RTX 6000 GPUs were used.

### A.4.2 Image classification

In the unguided regime, we evaluated the Counter-Hebb learning on the task of image classification and compared it with backpropagation and other biologically plausible learning algorithms under the same settings. The following results extend the results shown in Table 1 by evaluating the results obtained when learning with the MSE loss, in addition to the Cross-entropy loss reported in the main text. Note that there are two variations of CIM [Bozkurt et al., 2024], we report here the highest score obtained among the CIM experiments.

We repeat the same experiments as conducted in Bozkurt et al. [2024], our BU network employs a two-layer fully connected network, with a hidden layer of size 500 for both MNIST and Fashion-MNIST datasets and size 1,000 for CIFAR10. The standard Adam optimizer [Ruder, 2016] was used to optimize both the Cross-Entropy loss and MSE loss without any regularization. We trained for 50 epochs with an exponential learning rate decay with $\gamma = 0.95$. The initial learning rate was $10^{-4}$, and the batch size 20. All hyper-parameters but the initial learning rate were taken from the baseline experiments and were not optimized. The initial lr was selected from $1 \cdot 10^{-3}, 5 \cdot 10^{-4}, 1 \cdot 10^{-4}$ according to the best test results on the CIFAR dataset The results are presented in Table 4.

The results, shown in table 4, empirically validate that CH learning is equivalent to BP in the symmetric case, and approximates BP in the asymmetric case for both Cross-Entropy and MSE loss. Furthermore, the proposed asymmetric CH learning method shows a significantly smaller gap from BP compared with the other biological learning methods.

### A.4.3 Image classification with convolutional networks

The above experiments show that asymmetric CH learning approximates backpropagation well for fully connected networks (Multi-Layer Perceptron). Since biologically plausible learning methods often struggle to scale to larger networks and other types of architectures such as convolutional

networks [Ernoult et al., 2022], we conducted additional experiments to assess the robustness of CH in other settings. We examine the effects of varying the number of channels in each convolution layer and adding a weight decay regularization term on performance. All other settings remain the same, with only one parameter changed at a time.

For these experiments, we used the exact architecture and most hyper-parameters that were chosen for the Multi-MNIST benchmark. We ran two-layer convolutional networks on the MNIST and CIFAR10 benchmarks. Unlike the guided experiments, conducted on the Multi-MNIST benchmark, in these experiments, we used 32 channels for the convolution layers and increased the batch size to 256 for both MNIST and CIFAR10. The models were trained for 100 epochs, although most converged much faster.

The results shown in Figures 3, 4, 5, 6, compare different weight decay values and show that CH learning in the asymmetric case approximates the symmetric case (which is equivalent to backpropagation). Furthermore, in the backpropagation case, we observed a significant degradation in performance when increasing the weight decay term, up to the level where the model is not learning and the performances are near chance. Surprisingly, the asymmetric case is much more robust to this effect.

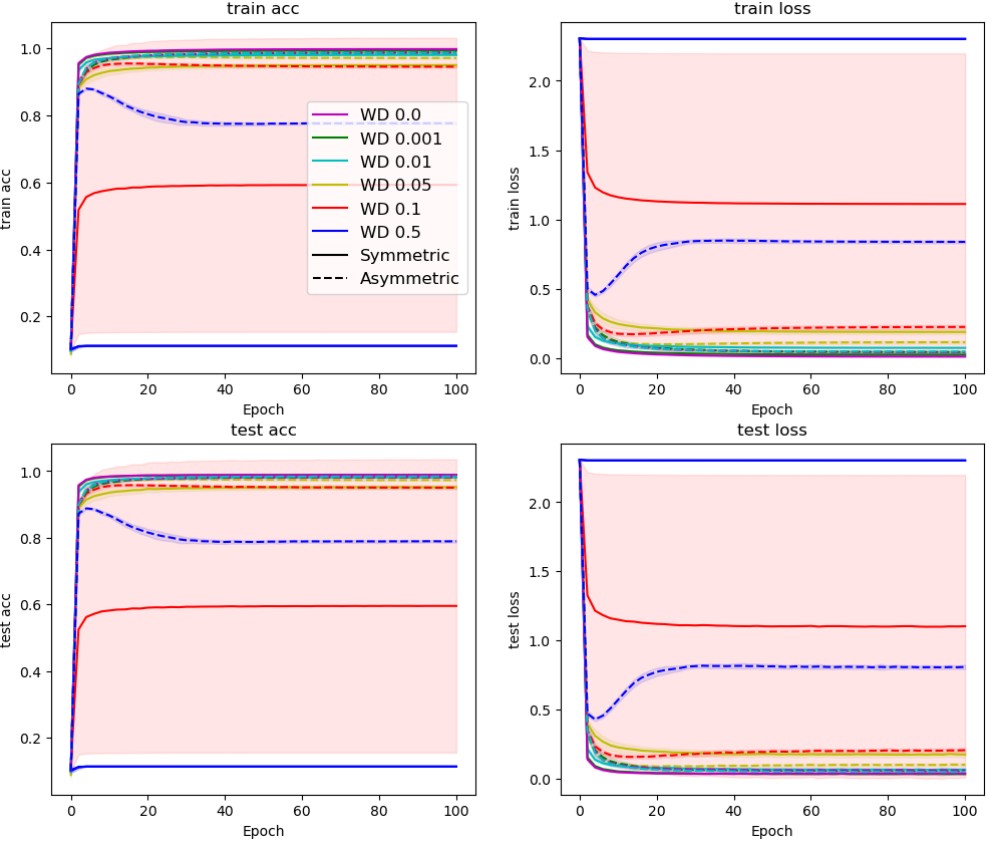

Figure 3: MNIST results: comparing different weight decay values and presenting the mean performance including std per training epoch averaged across 5 runs.

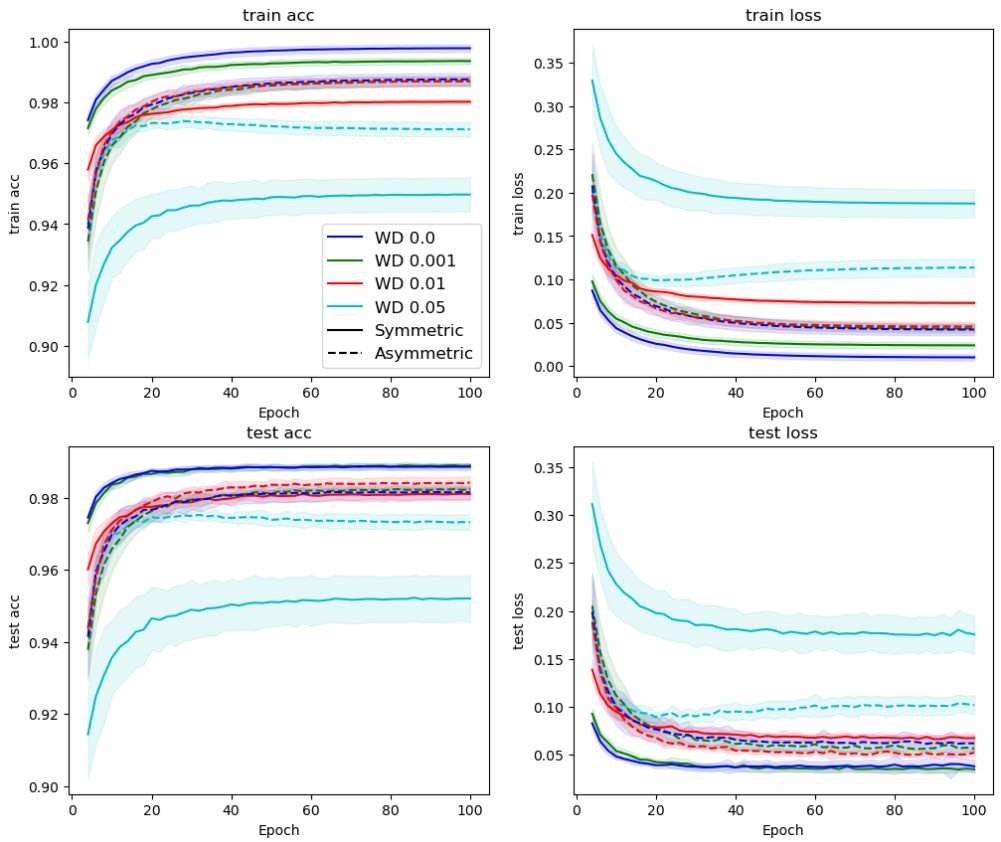

Figure 4: MNIST results: comparing different weight decay values and presenting the mean performance including std per training epoch averaged across 5 runs. Focusing on less weight decay factors, and starting from the 4th iteration for better visualization of the differences

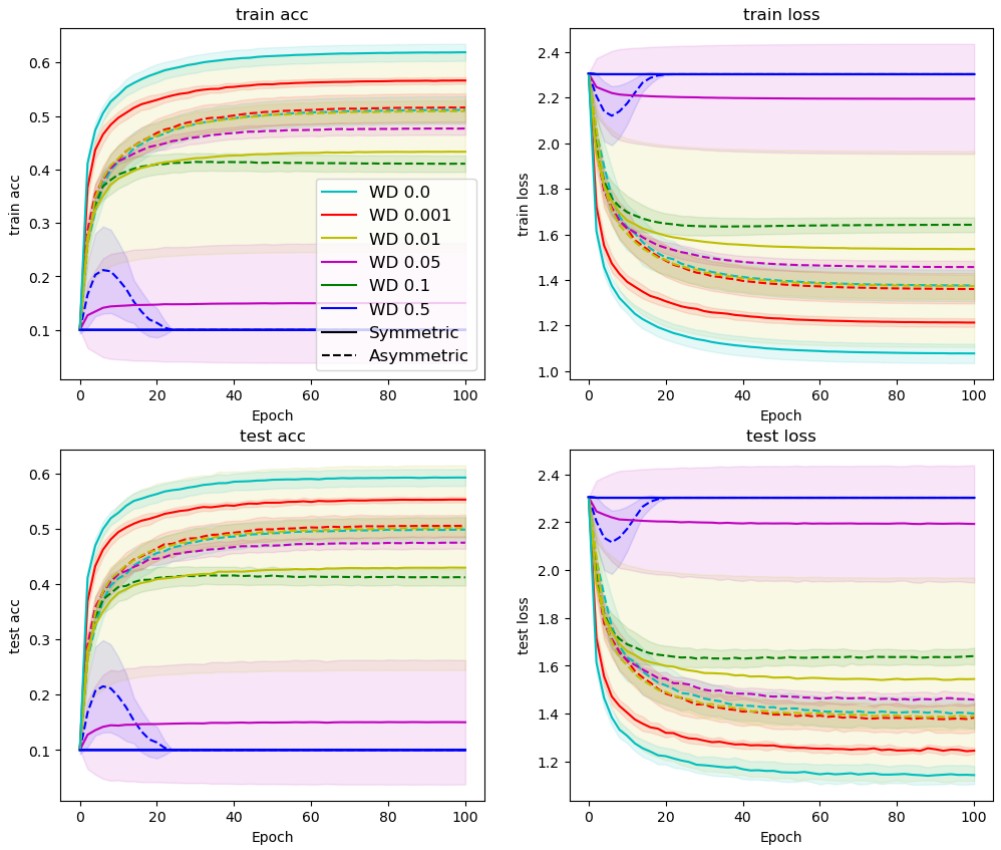

Figure 5: CIFAR10 results: comparing different weight decay values and presenting the mean performance including std per training epoch averaged across 5 runs.

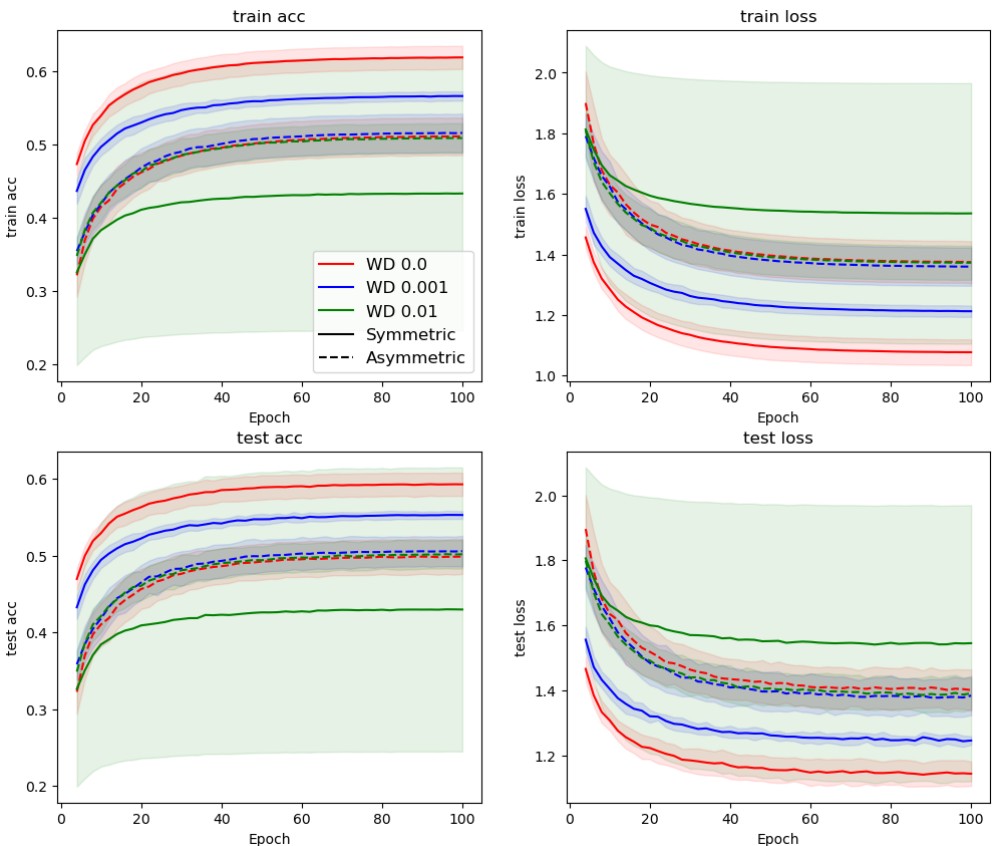

Figure 6: CIFAR10 results: comparing different weight decay values and presenting the mean performance including std per training epoch averaged across 5 runs. Focusing on less weight decay factors, and starting from the 4th iteration for better visualization of the differences

The results shown in Figures 7, 8, 9, compare different numbers of channels for each convolution layer. Similar to prior works [Ernoult et al., 2022], we observe that the gap between the symmetric and non-symmetric case increases as the capacity of the network increases, indicating that in large-scale tasks, backpropagation performs better than the asymmetric case.

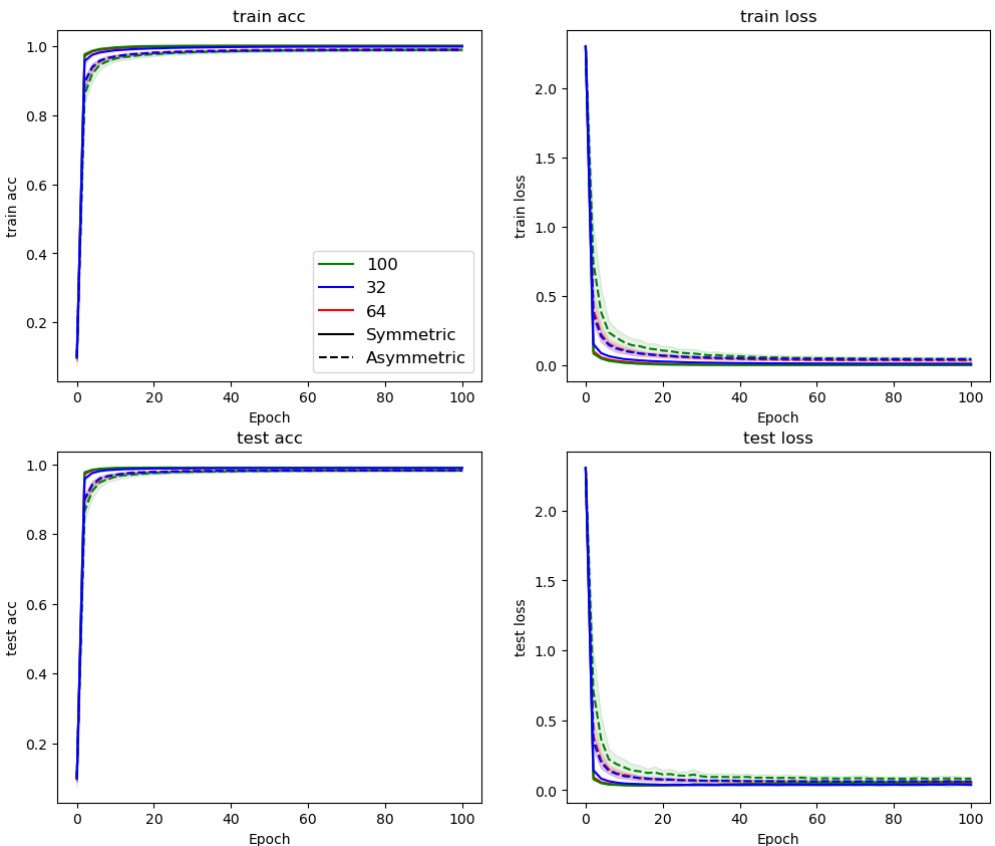

Figure 7: MNIST results: comparing different channels and presenting the mean performance including std per training epoch averaged across 5 runs.

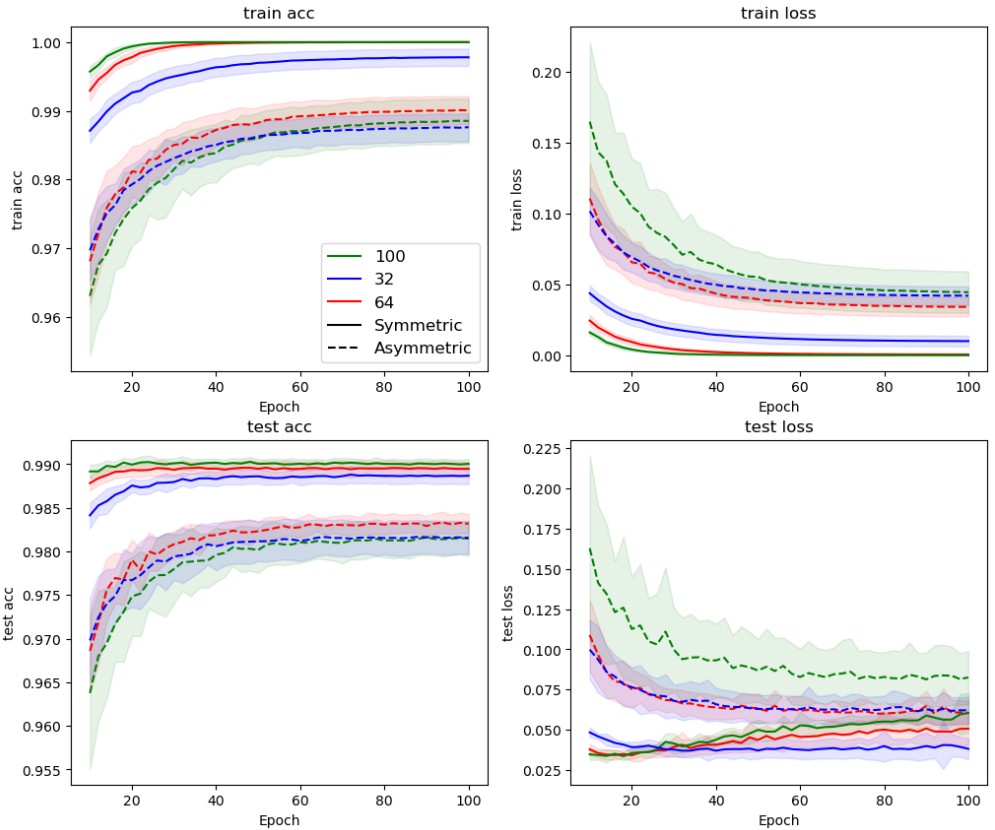

Figure 8: MNIST results: comparing different channels and presenting the mean performance including std per training epoch averaged across 5 runs. Starting from the 10th iteration for better visualization of the differences

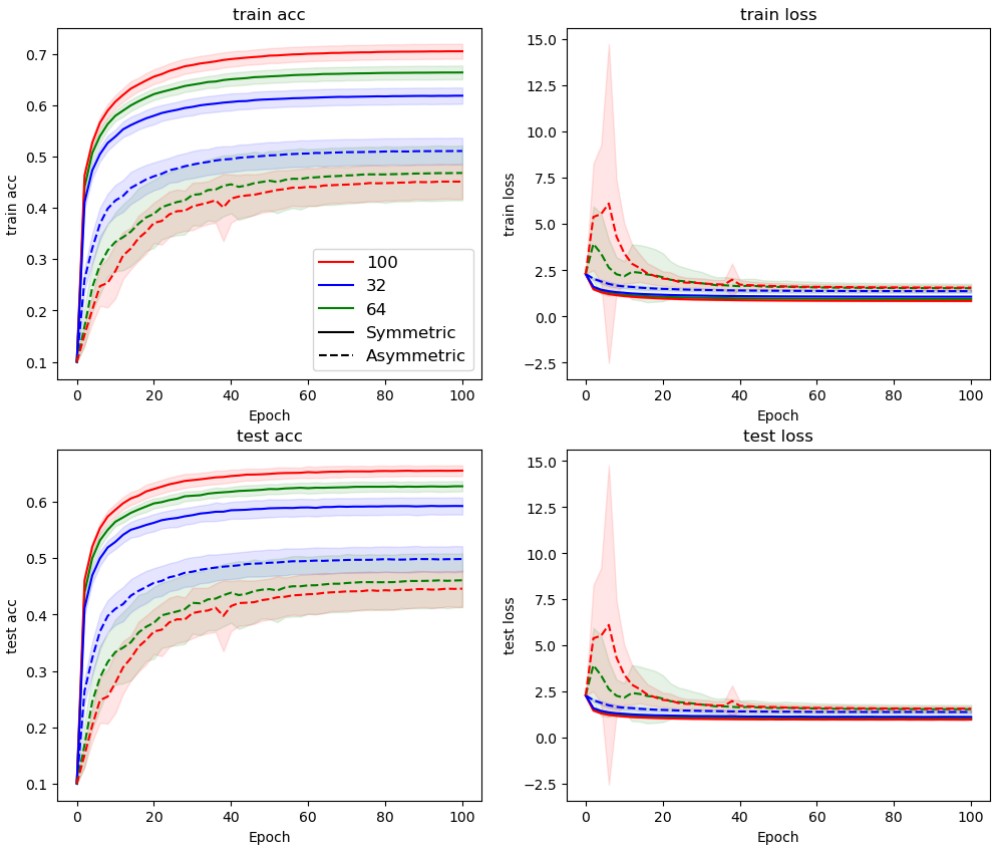

Figure 9: CIFAR results: comparing different channels and presenting the mean performance including std per training epoch across 5 runs.

### A.4.4 Guided visual processing settings

In the guided experiments, we evaluate our model on Multi-MNIST and CelebA, two common multi-task learning (MTL) benchmarks. Since current biological methods are not capable of guided processing, we compare CH with state-of-the-art non-biological optimization methods as reported by Kurin et al. [2022], replicating their setup and use their reported results.

However, in contrast to the baseline multi-task learning methods, we do not use any learning 'tricks' such as dropout layers, regularization, or special optimizers. Instead, our model is trained straightforwardly using the Adam optimizer [Ruder, 2016]. Another distinction from the baseline methods is that we do not use a validation set. As a result, we do not use an early stopping mechanism, and report the final results obtained from the last epoch which might introduce some noisiness. Furthermore, all hyper-parameters were only lightly tuned based solely on the training set.

### A.4.5 Multi-MNIST

Similar to the baseline experiments conducted in [Kurin et al., 2022], our BU network employs a simple architecture composed of 2 convolutional layers followed by a single fully-connected layer and ReLU non-linearity, along with an additional fully-connected layer as the decoder. Each convolution layer includes 100 channels, and a $5 \times 5$ kernel (a single stride and no padding). Similar to the baseline, the last fully connected layer size is 50. Additionally, to support the BU-TD structure, we replace all max-pool layers with strided convolution layers. The strided convolution layers have $2 \times 2$ kernel size with a stride of 2 (similarly to the max pool operation).

The standard Adam optimizer [Ruder, 2016] was used to optimize the Cross-Entropy loss without any regularization, as opposed to the baseline. Similar to the compared methods, we trained for 100 epochs with an exponential learning rate decay with $\gamma = 0.95$. The initial learning rate was $5 \cdot 10^{-4}$,

and the batch size was 64. We have chosen the learning rate from $[0.005, 0.001, 0.0005, 0.0001]$ based on the convergence rate on the train set.

In the experiments below, we keep all the settings above and evaluate the impact of using a single decoder as well as varying the number of channels in each convolution layer on the performance. The results shown in Figures 10, 11 demonstrate the robustness of the instruction-based method in a vanilla setting (as opposed to the baselines compared in the main text).

We observe that increasing the capacity of the model (number of channels) (Fig 10) increases the performance. Interestingly, in the asymmetric case, further increasing the capacity beyond a certain point reduces the performance.

Furthermore, when the model uses a single decoder (Fig 11) for both tasks, similar performances are maintained. This is a very important finding that highlights the ability of the TD stream to guide the BU stream. When employing a single decoder, the same network is used for both tasks (there are no task-specific parameters), thus the model cannot rely on task-specific parameters to handle multiple tasks.

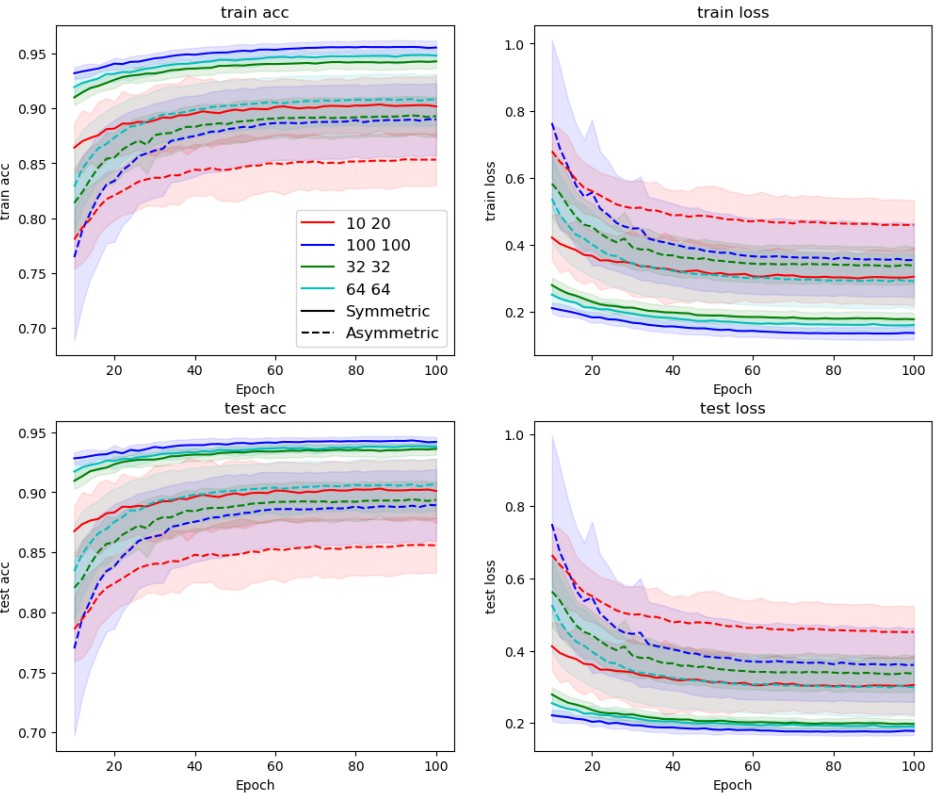

Figure 10: Multi-MNIST results: mean and std of the average task accuracy and loss per training epoch (starting from the 10th). Comparing different numbers of channels. On the left is the number of channels at the first convolution layer, while on the right is the number of channels at the second layer

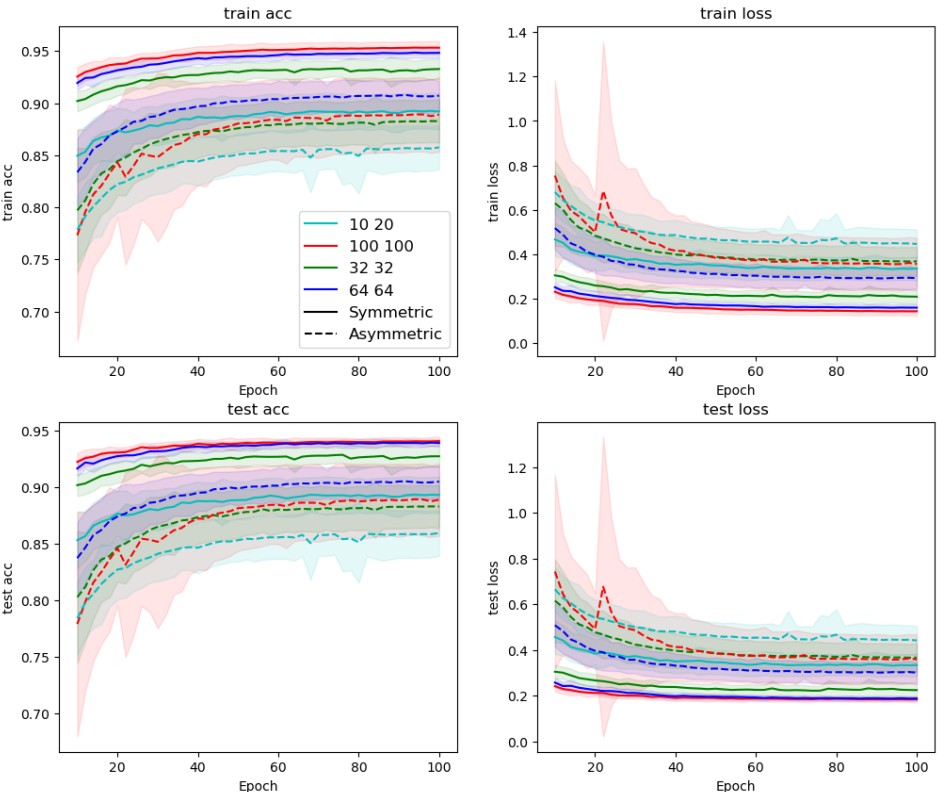

Figure 11: Multi-MNIST (single decoder) results: mean and std of the average task accuracy and loss per training epoch (starting from the 10th). Comparing different number of channels when the model uses a single decoder.

### A.4.6   CelebA

As done in previous work [Kurin et al., 2022], we employ a ResNet-18 [He et al., 2016] (without the final layer) with batch normalization layers [Ioffe and Szegedy, 2015], and the decoder is a single linear fully-connected layer with a single neuron output for binary classification. Additionally, we remove the last average pooling layer to support the symmetric BU-TD structure.

Batch normalization operates without reliance on learnable parameters, instead utilizing aggregated statistics such as the mean activation value of neurons across multiple iterations. Consequently, we implement distinct batch normalization for the BU and TD networks, with each network gathering statistics relevant to its own operations.

The standard Adam optimizer [Ruder, 2016] was used to optimize the Binary-Cross-Entropy loss without any regularization. Similar to the compared methods, we trained for $50$ epochs with an exponential learning rate decay with $\gamma = 0.95$. The initial learning rate was $5 \cdot 10^{-4}$ which is chosen from $[0.005, 0.001, 0.0005, 0.0001]$ based on the convergence rate on the train set. The batch size is slightly smaller than the baselines in order to fit the GPU memory, and is set to $100$ when the BU and TD networks share the same set of weights, and $64$ otherwise.

Statistics of the average task test accuracy obtained from the last epoch (no early stopping) with $5$ repetitions are reported in table 5. We compare the BU-TD model across multiple configurations, as described in Appendix A.4. In addition, we plot the test results during the training process, sampled every 5 epochs, in Figures 12, 13.

| Method | CelebA Test Accuracy |
|---|---|
| symmetric weights | $89.51 \pm 0.21$ |
| multi-decoders | $89.69 \pm 0.12$ |
| asymmetric weights | $79.25 \pm 1.63$ |

Table 5: CelebA results: mean and 95% confidence interval of the avg. task test accuracy (in percentages) across 5 runs.

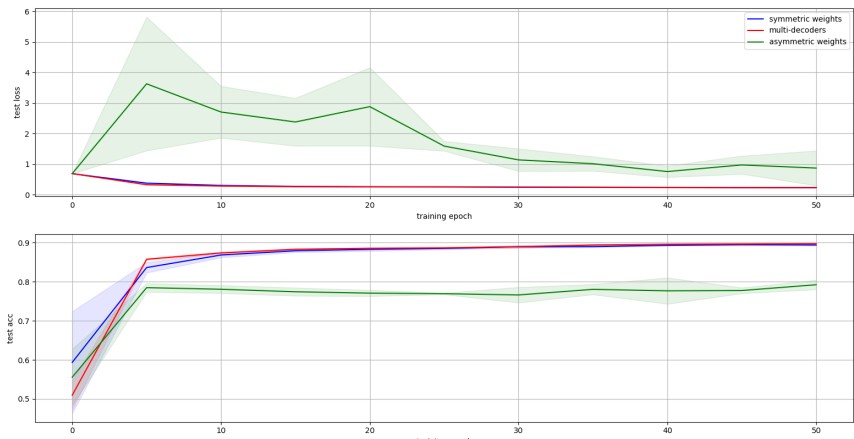

Figure 12: CelebA results: mean and std of the average task accuracy and loss on the test set per training epoch (sampled every 5 epochs).

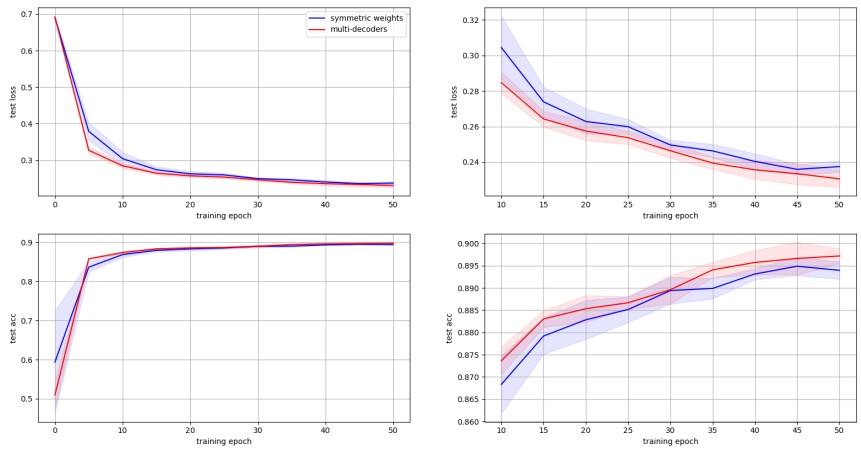

Figure 13: CelebA results: mean and std of the average task accuracy and loss on the test set per training epoch (sampled every 5 epochs). Within this figure, we omit the results of the asymmetric model due to being far from the other models. This allows for clearer observation of the distinctions among the remaining models. Additionally, on the right, presented the results starting from the 10th epoch.

The results on the CelebA dataset, which is more challenging are consistent with the Multi-MNIST results, demonstrating the BU-TD model's ability to scale and solve complex tasks through Counter-Hebb learning.

## A.5  Asymmetric weights

Weight symmetry poses a significant challenge in biological learning models, as copying weights across different locations is unrealistic in the brain, referred to as the 'weight transport' problem Grossberg [1987]. Therefore, unlike backpropagation, biological models use different weights for feedforward and feedback streams.

The experiments above showed increasing performance gaps with asymmetric weights compared to symmetric weights as task and model complexity increase. This phenomenon aligns with other biologically inspired methods. In this section, we further explore the effect of deviation weight symmetry on the model performance, focusing on both symmetry in the initialization of the weights, and symmetry in the subsequent updates. We use the terms 'symmetric model' for models with symmetric weight initialization and symmetric updates, 'asymmetric models' for weights initialized far from symmetry and symmetric updates. Additionally we introduced a new 'weak symmetric' scenario for models initialized symmetrically (or nearly symmetric), but are subject to noisy, asymmetric updates. For convenience, an overview of all scenarios is described in Table 6.

In the proposed weak symmetry scenario, the BU and TD weights are initialized symmetrically (or nearly symmetric), but we introduce noise to the Counter-Hebb update, simulating a more realistic case of noisy update where the BU and TD weight adjustments are not identical. Hence, the weights do not maintain symmetry during the learning. Specifically, at each update step, the update value of each weight is multiplied by a random (can be relatively large) noise from a $\mathcal{N}(1, \sigma)$ distribution, with a different random variable for each weight. It is worth noting that weak symmetry can also simulate a scenario in which weights are initialized asymmetrically with close to zero values, and by the time we start learning a task, the brain has already undergone some weight updates.

The results shown in Figures 14, 15 compare different magnitudes of noise applied to the Counter-Hebb update. It is shown that a weak symmetry is sufficient for achieving a similar performance as backpropagation (referred to as a 0 noise in the figures), even with a relatively large magnitude of noise. Since as the training progresses the results gradually become less symmetric, it demonstrates the ability of CH learning to converge to good solutions even when the BU and TD weights are not symmetric and have different values.

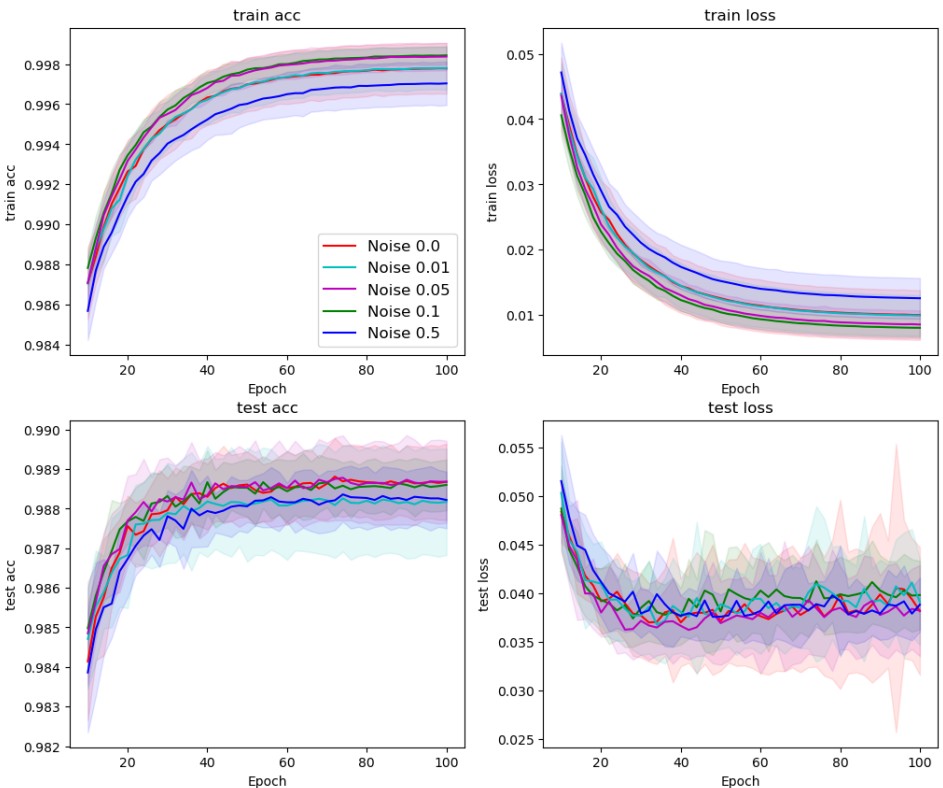

Figure 14: MNIST results: comparing different magnitudes of noise and presenting the mean performance including std per training epoch averaged across 5 runs. Starting from the 10th iteration for better visualization of the differences

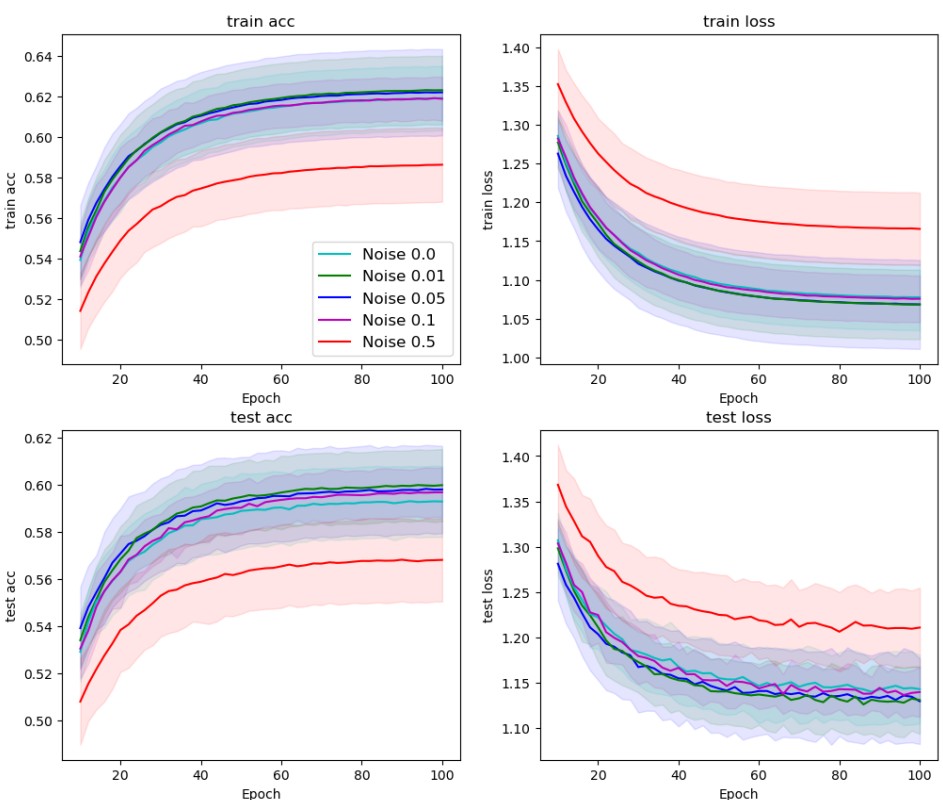

Figure 15: CIFAR results: comparing different magnitudes of noise and presenting the mean performance including std per training epoch averaged across 5 runs. Starting from the 10th iteration for better visualization of the differences

In addition, we evaluate the weak symmetric scenario in the guided-vision settings on the Multi-MNIST dataset. The results shown in Fig 16 demonstrate that an exact symmetry is not required by our model to perform similarly to backpropagation on a guided vision task, achieving similar performance despite using the TD network for both feedback propagation and guiding attention. The weak symmetry experiments achieve competitive (and even slightly higher) performances compared to exact symmetric weights. The dashed line indicates the final performance of the symmetric case.

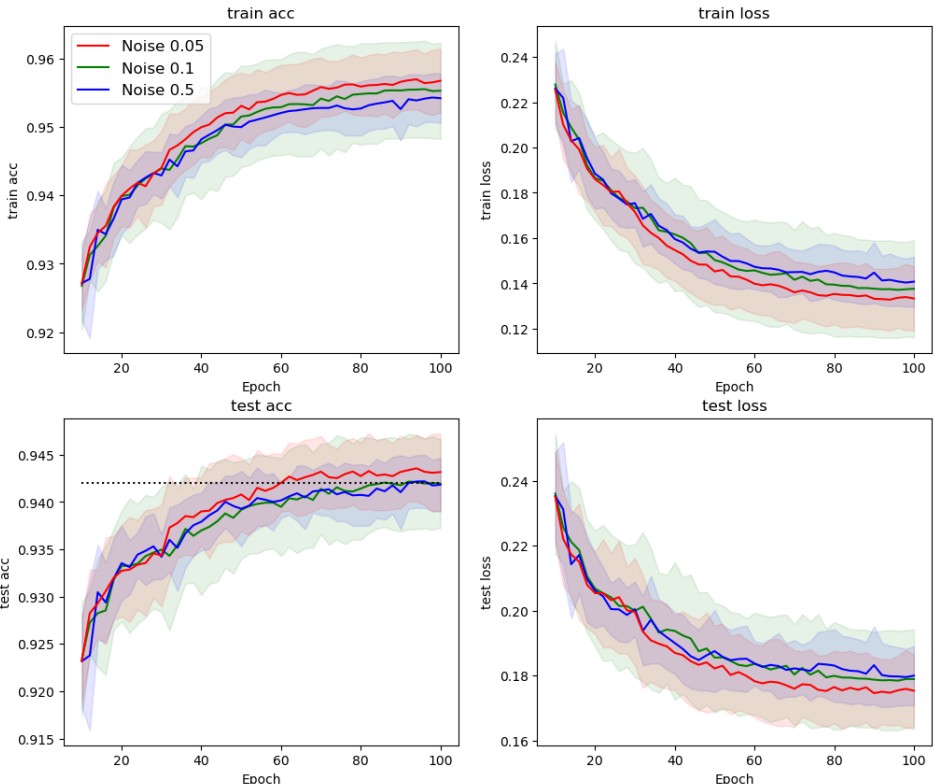

Figure 16: Multi-MNIST (weak symmetry) results: mean and std of the average task accuracy and loss per training epoch (starting from the 10th). Comparing different magnitudes of noise in the weak symmetry case. The dashed line indicates the final performance of the symmetric case.

To investigate the effect of deviation from weight symmetry on a larger model, we further compare the different weight asymmetry settings using the same ResNet18-like architecture from the CelebA experiments. Specifically, we evaluate Counter-Hebb learning on the CIFAR10 dataset under the following scenarios:

- symmetric

- asymmetric

- asymmetric with a weight decay term with magnitude of $10^{-2}$ and $10^{-3}$

- weak symmetric weights (noise std of $0.05$)

- noisy symmetric scenario which is similar to the weak symmetry but with noise applied to the weight initialization as well to start with asymmetric weights. Here, weights were initially set symmetrically, then each weight is multiplied by random noise with an std of $0.05$.

Additionally, we compare our model to the feedback alignment method [Lillicrap et al., 2016], as it achieved the highest performance among other biologically inspired methods in our non-guided experiments (Table 1). All models were trained for 50 epochs using the Cross Entropy loss, with a batch size was 128, learning rate of $10^{-4}$, and an exponential learning rate decay with $\gamma = 0.95$.

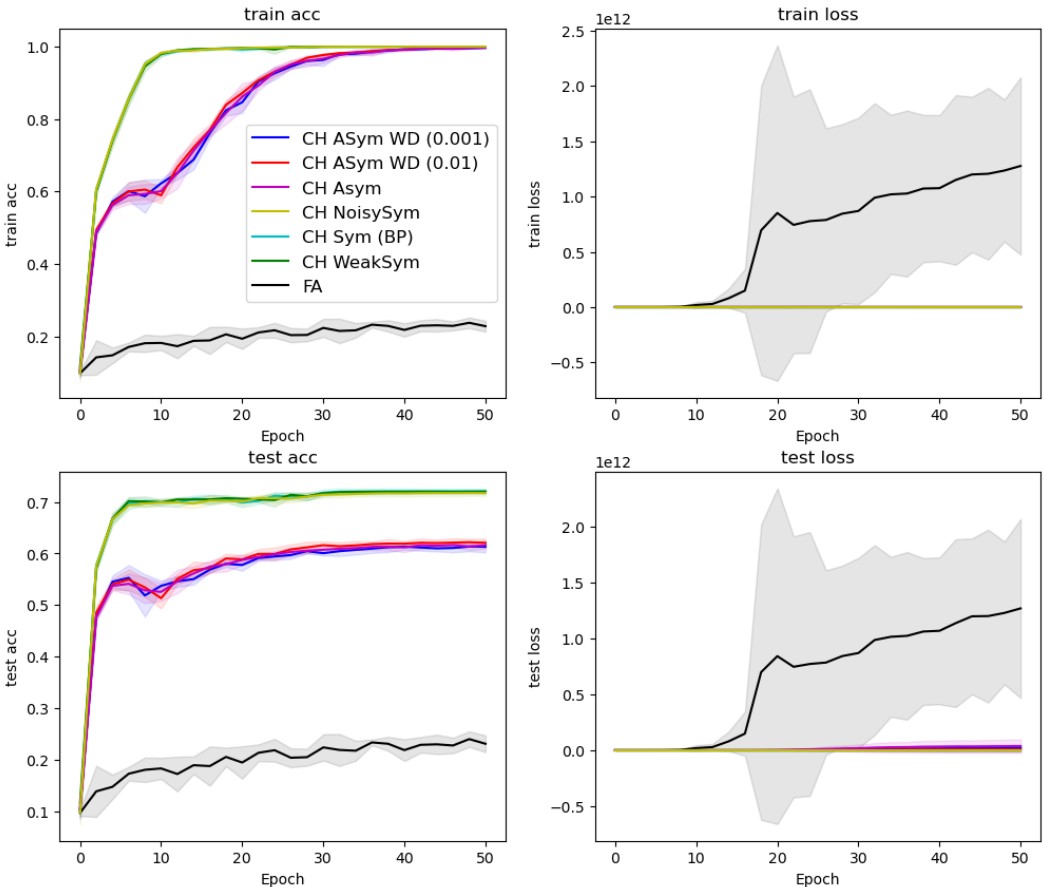

Figure 17: Comparing different weight symmetry using ResNet18 on CIFAR10 and presenting the mean accuracy including std per training epoch averaged across 5 runs.

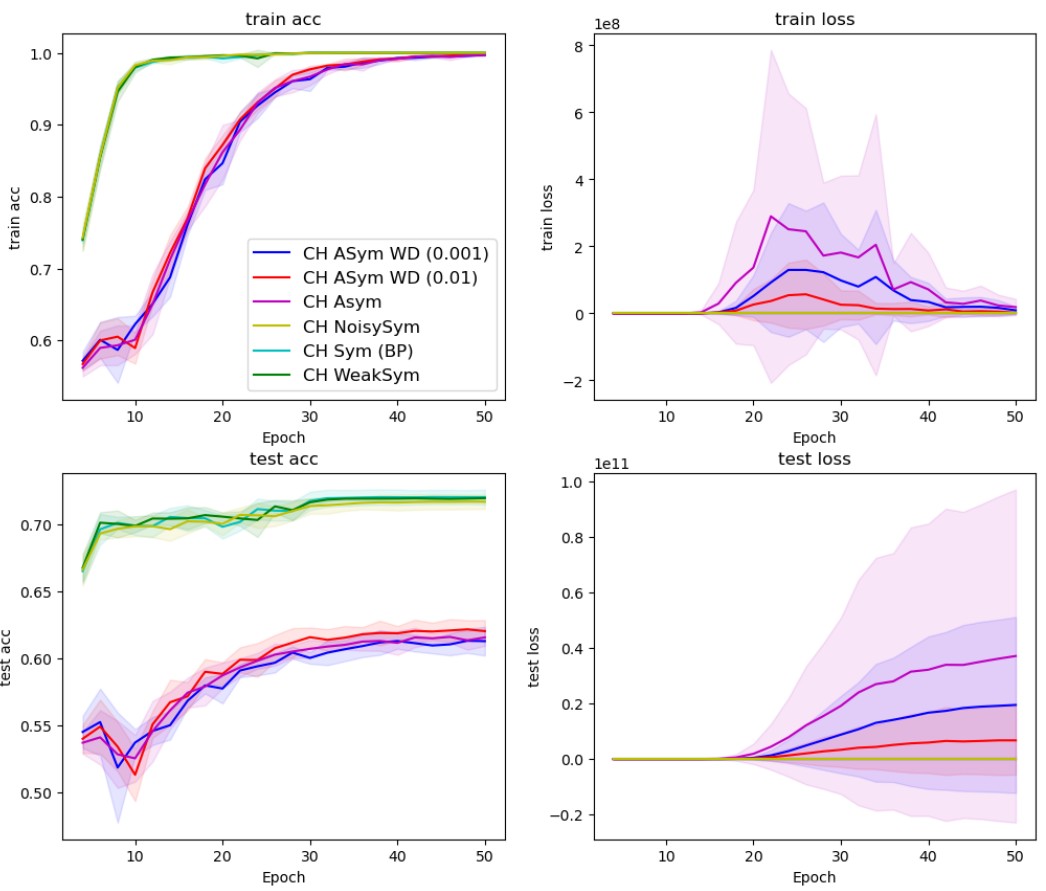

Figure 18: Comparing different weight symmetry using ResNet18 on CIFAR10 and presenting the mean accuracy including std per training epoch averaged across 5 runs. Excluding FA and starting from the 4th iteration for better visualization of the differences

The results presented in Figures 17, 18 show that initial weight symmetry is more critical for approximating backpropagation performance than gradually converging to symmetry later in the learning process. The asymmetric case is worse than the symmetric case both in terms of convergence speed and final results. Applying a weight decay term, which guarantees convergence to symmetric weights, does not appear to have a significant effect or improve results. The results also show that our model demonstrates significantly better scalability compared to the feedback alignment. In contrast, both in the weak symmetric case and the noisy symmetric case, where weights do not maintain symmetry due to noise, performance remains nearly identical to the symmetric case. These results indicate that exact weight symmetry is not necessary for backpropagation approximation.

### A.5.1 Asymmetric weights summary

The empirical experiments confirmed that symmetric Counter-Hebb learning is equivalent to learning with backpropagation. However, copying the same weights across different locations is unrealistic in the brain, referred to as the 'weight transport' problem Grossberg [1987]. Therefore, we explored the effect of deviation from weight symmetry on the model performance, focusing on both symmetry in the initialization of the weights, and symmetry in the subsequent updates. See Table 6 for an overview of these scenarios. In the asymmetric scenario, where weights are initialized far from symmetric but become more symmetric over training due to the symmetric updates, performance can approximate backpropagation on simple models and tasks. However, as task complexity and model size increase, performance begins to fall compared to backpropagation.

Table 6: Overview of the different weight symmetry settings used in our experiments. The columns indicate the name of the method, the state of the weights at initialization, intermediate weight states during training, and whether the final performance is similar to learning with backpropagation in order, respectively

| Method | Initialization | Intermediate | Performance |
|---|---|---|---|
| Symmetric | symmetric | symmetric | ✓ |
| Asymmetric | asymmetric | near symmetric | ✗ |
| Asymmetric + WD | asymmetric | symmetric | ✗ |
| Weak Symmetric | near symmetric | near symmetric | ✓ |

Adding a weight decay term guarantees convergence to weight symmetry, making the Counter-Hebb update approach the backpropagation update. Yet, the experiments show that late convergence to backpropagation alone does not ensure comparable performance. We hypothesize that, like backpropagation, Counter-Hebb and other biologically inspired learning models rely on proper weight initialization to achieve optimal results. By the time that symmetry of the weights is obtained, the weights may drift from their optimal initialization. This case will be similar to starting the standard backpropagation from non-optimal initialization conditions, leading in both cases to suboptimal performance.

In contrast, the weak symmetric and noisy symmetric scenarios, where weights remain close but do not converge to an exact symmetry, effectively approximate backpropagation and consistently maintain nearly identical performance. These findings suggest that exact weight symmetry is not essential for achieving performance comparable to backpropagation. Since Counter-Hebb naturally applies symmetric updates, performance appears more dependent on the initial weight configuration than on later symmetry convergence. Thus, discussions of biological feasibility in learning models may focus on achieving near-symmetric weight initialization, rather than starting with highly asymmetric weights and relying on convergence to symmetry.

### A.6 Sub-networks analysis

In our experiments, the model has exhibited the capability to solve multiple tasks by assigning a distinct task-specific sub-network for each task. In this section, we analyze the resulting sub-networks focusing on the Multi-MNIST experiment, where two tasks: "left" and "right" are involved. For the purpose of this analysis, we have evaluated a BU-TD model with symmetric weights and a single decoder. Our analysis shows the characteristics of the different sub-networks learned by the model and how they evolved during the learning process. Specifically, we have extracted for each task its corresponding sub-network every 3 epochs. Then we evaluated the size of the sub-networks and examined the level of overlap between them. The analysis is presented in Fig 19. The findings collectively provide insights into the learning dynamics of the model and its ability to develop task-specific representations.

From the figure, several findings can be drawn:

- Dynamic Nature of Sub-Networks: The sub-networks exhibit changes throughout the learning process, indicating that the model adapts and refines its sub-networks representations. This adaptation occurs especially in the earlier epochs of the training.

- Sparsity in Sub-Networks: A notable characteristic of the sub-networks is their sparsity (row 1)- a small percentage of active neurons. The percentage of active neurons drastically decreases at the early iterations until reaching a plateau. The level of sparsity is lower at the first layer as it represents the image signal and needs to capture a large number of pixels.

- Fixed top-level hidden layer: The top-level hidden layer is obtained by passing the task via the task head function. Since we do not update the task head during the training, This layer remains fixed during the learning.

- Similarity Between Sub-Networks: Despite the sparsity of the sub-networks, they demonstrate some degree of similarity (row 2). This outcome is likely due to the major correlation between the "left" and "right" tasks, as both tasks aim to identify one out of the same ten

digits. The first hidden layer (layer 0) exhibits a degree of similarity, likely arising from the overlap between the two digits. The following hidden layer (layer 1) shows the least similarity. The similarity then increases as moving deeper into the network. A possible explanation could be that early layers focus on the low-level image features at different image locations (left/right), while deeper layers focus on the high-level features for the classification of the identified digit.

The results show that the sub-networks are adjusted during the process, their separation can change at different layers and can depend on the similarity of the tasks.

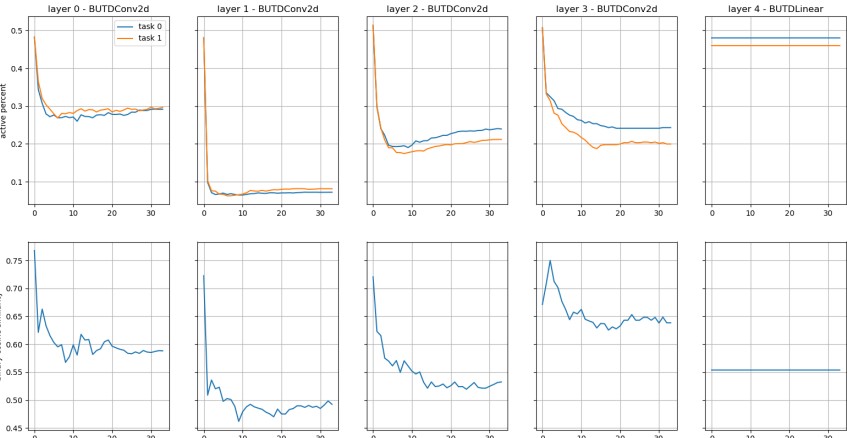

Figure 19: Analysis of the different sub-networks learned by the model and how they evolved during the learning process. We sampled the task-dependent sub-networks for each task Every 3 epochs during learning the Multi-MNIST data set. The columns in the figures represent the different hidden layers in the network, ordered from the first hidden layer on the left to the top hidden layer on the right. The X-axis of all figures represents the epochs during training. The first row shows the percentage of neurons that are being used in each sub-network for every layer. The second row shows the cosine similarity between the binary masking vectors of the two tasks, where 1 indicates an active neuron that is being used in the sub-network and 0 denotes an inactive one.

Therefore, the proposed method may offer some additional useful computational properties. In contrast to the compared baselines that require the full network for inference, our BU visual process is guided to operate only on a sparse sub-network, resulting in only a portion of the model that is used during inference. Consequently, after training, we can omit the unused parts of the model, resulting in a compact representation of the model, which is efficient both in terms of computation and memory. For example, we can drop approximately $80\%$ of the network when running the model in inference on Multi-MNIST. Furthermore, the compactness indicates the capacity of the model to accommodate a larger number of tasks within the same network.

### A.6.1 Functional sparse sub-networks

There has been a growing interest in the use of functional sparse sub-networks, following the Lottery-Ticket Hypothesis [Frankle and Carbin, 2018]. The hypothesis suggests that large dense networks contain smaller sub-networks that can be learned in isolation and match the performance of the full network on the learned task. This hypothesis has been supported by empirical evidence and was proven under certain conditions [Malach et al., 2020]. However, finding such sub-networks is challenging and is an active area of current research [Chen et al., 2021, Morcos et al., 2019, Ramanujan et al., 2020, Tanaka et al., 2020, Yu et al., 2022].

In this paper, we propose to extend this hypothesis suggesting that a sufficiently large network contains multiple overlapping sub-networks, each dedicated to a different task, resembling a sparse modular architecture. Our work suggests that these sub-networks can be naturally revealed by the same top-down mechanism used for propagating feedback signals in conventional networks (backpropagation). These findings, which are inspired by the observation of a unified top-down mechanism for both learning and guiding attention, highlight the potential benefits of the interactions between artificial intelligence and the human brain.

