# OpenReview forum: "Biologically Inspired Learning Model for Instructed Vision"
_NeurIPS.cc/2024/Conference — NeurIPS 2024 poster_

### Official Review · Reviewer_jamG · 2024-07-11

**Soundness:** 4
**Presentation:** 3
**Contribution:** 4
**Rating:** 8
**Confidence:** 4

**Summary:**

This paper proposes a novel and biologically plausible alternative to backpropagation. Their model is especially unique in how it incorporates visual attention through top-down processing mechanisms. Specifically, the model contains bottom-up (BU) and top-down (TD) networks that are symmetric to each other. The weights in the model are updated via a Counter Hebbian rule that strengthens the weight between a pre-synaptic and post-synaptic neuron when firing in the pre-synaptic neuron precedes firing in the neuron that is *counter* to the post-synaptic neuron (i.e., the neuron that is symmetric to, and laterally connected to, the post-synaptic neuron).

In Experiment 1, the authors demonstrate that the Counter Hebbian learning rule approximates backpropagation. In Experiment 2, the authors incorporate guided (or instruction-based) learning by adding an additional head (a one-hot representation of the relevant task) to the TD network. With this additional head, training now relies on 3 passes through the network. First, the task-head propagates task information downward. Second, the BU network (now guided by the task information from the first pass), processes the image to generate a prediction of the correct answer or label for the image. Third, the error-head propagates the error signal downward through the network.

Overall, the model is biologically motivated, incorporates top-down guidance that is unified (the same TD network is used for task information and error information), and presents a novel alternative to classic Hebbian learning. This is important because classic Hebbian learning presumes that a cell's firing will somehow propagate back to its dendrites to modify the synaptic weight, but this assumption is not necessary for Counter Hebbian learning. The use of the TD task-related head is important because visual guidance is a fundamental element of biological vision and is increasingly incorporated in AI models.

**Strengths:**

This is a great paper.

The model offers an attractive alternative to backpropagation that is both biologically inspired and reliant on local synaptic learning. (While the authors do not explicitly say this, it also offers a potential biological basis for why backpropagation methods work; i.e., because they approximate the Counter Hebbian learning described here.) The Counter Heddian learning procedure is novel and incorporates both BU and TD processes (which are critically biologically, but usually conflated in AI models).

The model offers a simple way to incorporate task-based instruction/attention into visual processing. I found it very clever that the model uses the same TD network for task-instruction and error propagation (just with different heads). It's both computationally elegant to use a single TD stream and more biologically realistic.

I only caught this in the Appendix, but the authors also note that their models do not rely on any computational "tricks" ("such as dropout layers, regularization, and special optimizers"). This is a major strength as well, even though it is not highlighted in the main manuscript.

Finally, the authors are able to explain a somewhat complex set of ideas (i.e., Counter Hebbian learning + TD processing + BU processing + multiple model passes) very clearly and simply.

**Weaknesses:**

These weaknesses are all minor constructive criticisms:
1) The references are formatted in a style that makes the paper very hard to read. Currently the references are formatted as Author [year], but it would be easier to parse the sentences if they were formatted as either bracketed numbers or as (Author, year).
2) There are some typos throughout (e.g., page 1 line 26, page 2 line 39, page 2 line 60, page 4 line 181, page 5 lines 190-192).
3) Some of the abbreviations are unnecessary (they require more working memory from the readers, but are only used 1 or 2 time). The specific abbreviations I noticed that were rarely actually used in text were BP, FA, and TP.
4) Some of the discussion of backpropagation in Related Work could be explained more clearly (specifically page 5 lines 94-98).
5) Figure 3 in the appendix is difficult to interpret (visually). The written explanation of the figure is quite clear, but the visual doesn't make it any clearer (and may actually make it slightly less clear).
6) The graphs in the appendix need axis labels to be interpretable (figures 4-16).

**Questions:**

Q1: How does this model compare--in terms of scalability--to traditional backpropagation methods? Does this model take much more compute time and/or resources than the state-of-the-art methods they tested?

Q2: Do the authors believe that this model could be extended for use in unsupervised/self-supervised learning? In biological visual systems, visual attention is also guided by saliency and not just task demands. Could future versions of the model incorporate saliency-based attention as well? To be clear, the answer to this question would not change my opinion, but I'm curious about their thoughts on it.

**Limitations:**

Yes, the authors have adequately addressed limitations.

---

> ### Author Rebuttal · Authors · 2024-08-06
>
> Thank you for your comprehensive and detailed review of our paper. Your feedback will be used, and we will update the final version of the paper according to your comments.
>
> > W1
>
> Thank you for the constructive comment; we will change the format to make it easy to parse.
>
> > W2
>
> Thank you for the catch, we will fix all the typos in the final version.
>
> > W3
>
> Thanks, we will remove unnecessary abbreviations in the final version (e.g. FA and TP).
>
> > W4
>
> Thanks, we will update the final version to clarify this discussion. We will be glad to provide the revised paragraph in further correspondence.
>
> > W5
>
> Following your comment, we think that it will be better to just use the text (possibly with minor changes) and remove the figure. The reason for the figure was to visualize the possibility of providing two different inputs in the same top-down stream: an error signal for BP and task-instruction for guided vision.
>
> > W6
>
> Thank you for pointing it, the x axis represents the epochs, and the y axis corresponds with the title of the sub-figure (e.g. ‘train loss’). We will add axis labels and increase the size of the graphs so they will be more interpretable in the final version.
>
> > Q1
>
> We considered a couple of scenarios, and we think that the overall requirements in compute time and resources will be similar to traditional backpropagation, since both use a single forward (bottom-up) pass and a single backward (top-down) pass. In terms of memory, CH requires storing two sets of weights, for the BU and TD networks, but in a computer simulation it is possible to enforce symmetric weights and have similar memory requirements. We conducted an experiment for empirical comparison, training a ResNet18 architecture on CIFAR, an unguided classification task (trained on a single NVIDIA A10).
>
> We compared the memory requirement and training time (50 epochs), showing below the results (in the format of (minutes : seconds)).  CH Sym uses a single set of model weights and CH Asym uses two sets.
>
> | Method | Time | Memory |
> |-----------|-----------|-----------|
> | BP | 9:11 | 1,122 MB |
> | CH Sym | 9:00 | 1,650 MB |
> | CH Asym | 9:25 | 1,832 MB |
>
> The results show that the computation time is roughly the same for all methods, but the memory requirement is slightly higher for asymmetric CH. Note that the low bp memory is due to the highly optimized Pytorch implementation of backpropagation.
>
> The example above is an unguided classification task. In the guided settings, the comparison becomes more complex because there is currently no standard model for guided vision. In general, in order to perform guided vision, a model needs to process the guidance information, and then integrate the results with the visual processing. To learn such a model with traditional backpropagation, a backward pass needs to propagate throughout the entire process including both the guidance and visual processing. Our model suggests an efficient scheme of the guided scenario since it reuses the same TD stream for both the error propagation and the instruction processing, reducing the complexity and memory requirement of the model.
>
> > Q2
>
> We think that it will be possible to incorporate bottom-up saliency-based attention in the model, and also to create a useful integration of bottom-up saliency with top-down attention. There has been some work on combining BU saliency with TD attention, for example [1], which showed in fMRI studies evidence for additive combination of BU saliency and TD attention in early visual areas, and another [2] found fMRI evidence for combined effects of BU saliency and TD attention specifically in visual area V4.
>
> It seems that our proposed model can naturally combine the two at multiple levels, and not necessarily in an additive manner, but also in other ways, which can be under TD control. In our model, BU saliency signals will propagate along the BU stream, and at the same time TD task signals will propagate along the TD streams. The cross-stream connections can allow, for example, to guide the top-down instruction signals towards high-salience locations, e.g. to segment or classify specifically regions of high saliency. In the opposite direction, TD signals could modify the saliency map, in a way that depends on the task instruction, rather than using a fixed additive combination. It could be interesting to explore such possibilities in the future and compare the model with the available empirical evidence.
>
> [1] Poltoratski S, Ling S, McCormack D, Tong F. Characterizing the effects of feature salience and top-down attention in the early visual system. J Neurophysiol. 2017, 118(1):564-573.
>
> [2] Melloni L, van Leeuwen S, Alink A, Müller NG. Interaction between bottom-up saliency and top-down control: how saliency maps are created in the human brain. Cereb Cortex. 2012, (12):2943-52.

---

> > ### Author Response · Authors · 2024-08-12
> >
> > Thank you again for your valuable feedback. In response to your comments on the related work, we have revised the text to improve clarity. Below is the updated version:
> >
> > The Predictive Coding approach suggests that in visual processing, feedback connections carry predictions of neural activities, whereas feedforward streams carry the residual errors between these predictions and the actual neural activities [1]. It has been demonstrated that when predictive coding is used to train a neural network in a supervised learning setting, it can produce parameter updates that approximate those computed by backpropagation [2, 3]. These results have been further developed under additional assumptions, leading to predictive coding variants that produce the exact same parameter updates as backpropagation [4, 5]. However, the modifications necessary for these methods to approximate or be equivalent to backpropagation are criticized for reducing their biological plausibility [6, 7].
> >
> > [1] Rao, Rajesh PN, and Dana H. Ballard. "Predictive coding in the visual cortex: a functional interpretation of some extra-classical receptive-field effects." Nature neuroscience 2.1 (1999): 79-87.
> >
> > [2] Whittington, James CR, and Rafal Bogacz. "An approximation of the error backpropagation algorithm in a predictive coding network with local hebbian synaptic plasticity." Neural computation 29.5 (2017): 1229-1262.
> >
> > [3] Millidge, Beren, Alexander Tschantz, and Christopher L. Buckley. "Predictive coding approximates backprop along arbitrary computation graphs." Neural Computation 34.6 (2022): 1329-1368.
> >
> > [4] Song, Yuhang, et al. "Can the brain do backpropagation?---exact implementation of backpropagation in predictive coding networks." Advances in neural information processing systems 33 (2020): 22566-22579.
> >
> > [5] Salvatori, Tommaso, et al. "Reverse differentiation via predictive coding." Proceedings of the AAAI Conference on Artificial Intelligence. Vol. 36. No. 7. 2022.
> >
> > [6] Rosenbaum, Robert. "On the relationship between predictive coding and backpropagation." Plos one 17.3 (2022): e0266102.
> >
> > [7] Golkar, Siavash, et al. "Constrained predictive coding as a biologically plausible model of the cortical hierarchy." Advances in Neural Information Processing Systems 35 (2022): 14155-14169.

---

> > ### Comment · Reviewer_jamG · 2024-08-12
> >
> > Thank you for your thoughtful engagement with my comments and questions!
> >
> > I remain quite enthusiastic about this work. My score was already high (8), and I don't think the work justifies a higher score (9, which would imply it is "groundbreaking"), but I do continue to advocate for its acceptance.

---

### Official Review · Reviewer_MScR · 2024-07-13

**Soundness:** 3
**Presentation:** 3
**Contribution:** 3
**Rating:** 6
**Confidence:** 2

**Summary:**

This manuscript proposed the first biologically motivated learning model for instructed visual models that integrates bottom-up and top-down pathways mimicking the visual cortex. The model employs the TD pathway for both guiding attention and propagating signals. Experiments demonstrate its capability across multiple tasks and achieve competitive performance with current AI models.

**Strengths:**

The paper is generally well-organized with a solid theoretical foundation.

**Weaknesses:**

NA

**Questions:**

NA

**Limitations:**

The potential limitations have been thoroughly discussed.

---

> ### Author Rebuttal · Authors · 2024-08-06
>
> Thank you for your review and your succinct and precise summary. We welcome any comments you have on the additional discussion in the global rebuttal.

---

### Official Review · Reviewer_LFXA · 2024-07-13

**Soundness:** 2
**Presentation:** 2
**Contribution:** 2
**Rating:** 3
**Confidence:** 3

**Summary:**

This work focused on proposing an efficient biologically plausible learning framework. It is inspired from the cortical-like combination of bottom up and top down processing. The top-down part provide both guidance for visual process and feedback signals for learning part. It further introduce a "Counter-Hebb" mechanism to modify synaptic weights.

**Strengths:**

**Method**

To mimic the biological network with both top-down and bottom-up stream in the artificial neural network is an interesting direction.  The proposed "Counter-Hebb" mechanism is novel and effective.

**Evaluation**

The framework is demonstrated on multiple benchmarks on image classification and multi-task learning, achieving comparable results compared to existing biologically plausible learning rules.

**Weaknesses:**

**Novelty**

The main framework utilize the both top-down and bottom-up stream for guiding visual process and learning. The core idea is similar to predictive coding, where the novelty is limited. Discussing the fundamental differences with existing approaches will be helpful.

**Biologically plausibility**

The  method has restricted TD and BU to be symmetric, "same connectivity structure", which violates the biologically plausibility.

**Benchmark**

The methods is only evaluated on toy-level image classification tasks, and does not outperform existing approaches.

**Connection with VLMs**

This paper mentions connections with VLMs, while its relation is unclear, and there is no demonstration and evaluation performed in the studies.

**Questions:**

1. How is the proposed method related to VLMs?

2. How is the proposed method related to existing biologically plausible approaches (predictive coding, feedback alignment)?

3. Why $\bar{f_l}$ needs to have the same connectivity structure as $f_l$?

4. Why average pooling is excluded in the framework.

**Limitations:**

No potential negative societal impact.

---

> ### Author Rebuttal · Authors · 2024-08-06
>
> Thank you for your valuable feedback. We appreciate your interest and constructive comments.
>
> > W1 + Q2
>
> Predictive coding is among the best biologically plausible models for learning from errors and for representation learning. We discuss some differences and novel aspects in the biological mechanisms of Counter-Hebb learning, but a major contribution of our paper is the integration of guided vision, through top-down attention, in the same TD stream used for learning. Combining these two major functions is important both for modeling the biological TD stream, as well as for improving visual processing from a computational standpoint. For more information on the role of top-down attention in human vision and computer vision, please refer to the global rebuttal.
>
> > W2
>
> The knowledge about the structure of the BU and TD streams is incomplete, but the overall scheme is essentially consistent with the main aspects of the model we used. We show a schematic figure (attached in the global rebuttal) of the known connectivity between the BU and TD in primates, adapted from [1,2], showing how the basic counter-streams structure is embodied in cortical connectivity of the ventral stream. It shows the main connections between successive areas labeled I, II.  The BU path goes through layer 4 to layer 3B of area I and then to the next area. The TD path goes from layer 2/3A and layer 6 of area II to 2/3A and 6 of the lower area.
>
> Similar to our model, the cortical circuit has two streams, which are interconnected in both directions, between layers 4 and 6, and between the superficial layers 2, 3. Dashed arrows are pathways that skip a step in the stream. Although the precise connectivity at the level of individual neurons is not known, the symmetry between the BU and TD streams is obeyed in the circuitry at the population level: for the connections from layer 4 to layer 3B on the BU stream, there are connections in the opposite direction between their corresponding counter-neurons, from 2/3A to layer 6.
>
>
> > W3
>
> There are two comparisons, with biological and non-biological models. Regarding the scale of the model, in the biological comparisons, we address the CelebA benchmark using ResNet-18, which is large compared with other biological models. See also an additional experiment related to scale in the PDF attached to the global response.
>
> In terms of performance, as shown in Table 1, our method is among the top biological models, outperforming other baselines. With respect to non-biological, AI models, as mentioned in the main text, our main contribution is not outperforming existing AI approaches, but rather to show the ability of our model to achieve a similar level of performance compared with non-biological models, despite using the constraints of biological models, and despite using the same top-down stream for both learning and guiding attention.
>
> > W4 + Q1
>
> A notable similarity between our model and VLMs is the ability to perform guided vision using two parallel streams, one carrying visual information, and the other carries high-level, more cognitive information. In VLMs, the vision stream extracts visual information from visual inputs, while the language stream handles high-level semantic processing and integrates general knowledge from Large Language Models. This allows the language stream to take natural language instructions and direct the visual processing to focus on information that is relevant to the task. For example, in Visual Question Answering, the visual process is instructed to extract visual information that is relevant to the question.
>
> Our proposed method shares a similar concept. Unlike existing biological models of vision, which rely solely on visual inputs, our model incorporates both visual and instructional information during visual processing. Our model consists of two streams: a bottom-up stream for processing input images and a top-down stream for handling instructional information, which provides high-level semantic context. Similar to VLMs, the top-down stream guides the visual processing in the bottom-up stream to extract visual information relevant to the given task. For more details on the connection to VLMs, please refer to the global rebuttal.
>
> > Q3
>
> The connectivity structure is required in order to derive the mathematical results shown in the paper: the equivalence to backpropagation in the symmetric case, and the approximation of backpropagation in the asymmetric case. However, it could be interesting to explore in future research Counter-Hebb learning when relaxing this structure to symmetry at the population level of neurons.
>
> > Q4
>
> In ResNet architecture, global average pooling is used at the top core layers to down-sample the entire channel representation into a single neuron. While this operation is effective for classification tasks, it is less suitable for our framework, which focuses on instruction guided processing. In our guided vision learning algorithm, the first top-down pass is used for receiving and propagating instruction representations throughout the network. Utilizing global average pooling would restrict this process by compressing all instruction information into a single neuron per channel.
>
>
>
> [1] Markov, N. T. et al. (2013) ‘Cortical high-density counter-stream architectures’, Science, 342(6158).
>
> [2] S. Ullman. Sequence-seeking and counter streams: A computational model for bi-directional information flow in the visual cortex. Cerebral Cortex, 5(1) 1-11, 1995.

---

> > ### Comment · Reviewer_LFXA · 2024-08-13
> >
> > Thanks the authors for their additional clarifications. While I share similar concern as reviewer n2CE about the biological plausibility of symmetry connectivity between BU and TD paths. Meanwhile, I think the proposed method is lack of demonstrations in larger scale and more complicate tasks, as well as benchmarking with more advanced non-biological models, and the writing and overall presentation needs improvement. Therefore I maintained my original score.

---

> > > ### Author Response · Authors · 2024-08-13
> > >
> > > Thank you again for your valuable feedback that will help us improve the clarity of the paper.
> > >
> > > We add one additional issue that we think still merits consideration, since it relates to a basic aspect, which we realize from the comments from you and from reviewer n2CE that it was not clear in our presentation.
> > >
> > > What was not sufficiently clear is that we do have an asymmetric version that reaches performance essentially indistinguishable from back propagation. In our model we discussed two versions of weights asymmetry which we termed ‘asymmetry’ and ‘weak symmetry’ (or `weak asymmetry`), but the distinction was not clear; please refer also to the rebuttal response to reviewer n2CE (W3).
> > >
> > > The version that reached back-propagation performance is what we refer to as the ‘weak asymmetric’ case, where the two streams have different sets of weights, and the weights are updated with noisy, asymmetric updates. In this version the main requirement is to have in initialization, prior to training, sufficiently close to symmetric weights. We considered also a simple initialization scheme that satisfies this requirement. It can be obtained by weak initial weights, and an arbitrary activation of the entire network. Applying the Counter-Hebb under these conditions will automatically push the network toward this requirement.  Our experiments showed that this initialization, followed by noisy updates, results in model performance that remains very close to the standard backpropagation. Biologically, we suggest that the claustrum, which provides input to all cortical regions, could supply such activation. Following your comments, we will add this discussion to the paper.

---

### Official Review · Reviewer_n2CE · 2024-07-30

**Soundness:** 2
**Presentation:** 2
**Contribution:** 2
**Rating:** 6
**Confidence:** 3

**Summary:**

This study proposes a new biologically-motivated learning rule as an alternative to backpropagation. This “Counter Hebbian” rule separates the forward and backward path into two similar but non-identical pathways, called Bottom Up and Top Down by the authors, and allows the TD stream to gate the flow of information through the network. The authors run simple experiments on MNIST, Fashion MNIST, and CIFAR10, as well as on small multi-task datasets Multi-MNIST and CelebA.

**Strengths:**

The Counter Hebb rule is an exciting proposal, and this paper does a thorough job of motivating and explaining the algorithm.

**Weaknesses:**

1. It would be helpful to see additional ablation experiments that break apart the components of the proposed CH learning rule.

2. There are very few experiments in the main text of the paper. Most of the numbers presented in Tables 1 and 2 are from other papers (Bozkurt et al. 2024 and Kurin et al. 2022).

3. The asymmetric case is much more biologically plausible, however deep in the text the authors admit that it does worse than backpropagation (e.g. lines 667-668 in the Appendix). This dampens some of the strongly worded claims in the abstract and at the beginning of the paper, which states, for example “...achieving competitive performance compared with AI models on standard multi-task learning benchmarks,” even though this is only true for the symmetric case.

4. The model is incredibly small. It would be helpful to show that this CH algorithm works for more layers and at a larger scale. Feedback Alignment, for example, is known to scale poorly beyond a few layers. It is surprising that in 2024 the field of biologically plausible learning rules is reporting results for MNIST, when the rest of the field is tackling larger models such as Llama and much more difficult tasks.

**Questions:**

1. Is the CH algorithm applied to the convolutional layers as well in the Multi-MNIST case?

2. How do you address the symmetry of convolutional layers?

3. How should we interpret the results in Table 2? Is it fair to compare CH with SMTO approaches from Kurin et al.? Why not compare directly with backpropagation in the main text? (I see this is included in Table 3 of the Appendix)

**Limitations:**

* The figures in the appendix are incredibly small and difficult to read. This makes it difficult to have any confidence in the ablation studies
Line 312 “were carefully optimized over the years”. What does this mean?

* GaLU is almost similar to Gated Linear Unites (Shazeer 2018). It could be worth citing this paper or something similar

* It is not possible to see the “Asymmetric” results in Figure 14. Shouldn’t they be worse for Multi-MNIST?

* "Instructed Visual Processing" is somewhat of a vague term to use.

---

> ### Author Rebuttal · Authors · 2024-08-07
>
> Thank you, we appreciate the time and effort and the comprehensive and helpful comments.
>
> > W1 + W2
>
> Additional experiments are included in the appendix and are only briefly mentioned in the main text due to capacity limitations. These experiments examine various components of CH learning, such as guided and non-guided learning and different schemes of weight symmetry and asymmetry. They also assess the model's robustness to different architectures, loss functions, etc. Our findings indicate that combining BP with TD attention does not compromise performance. following your comments, we will stress this point in the main paper and we will add pointers in the main text to the relevant parts in the appendix
>
> > W3
>
> In analyzing weight symmetry, we looked at two aspects: symmetry in the initialization of the weights, and symmetry in the subsequent CH updates. We used 'asymmetry' or 'asymmetric initialization' for weights initialized far from symmetric, and 'weak symmetry' for symmetric (or nearly symmetric) initialization with asymmetric noisy updates. To avoid confusion, since both scenarios represent asymmetric weights, we will use the explicit term ‘asymmetric initialization’ in the revised version.
>
> Our experiments show that while asymmetric initialization is sometimes worse than BP, weak symmetry consistently yields results similar to BP [see App Fig 10, 11, 14, and Fig 1 in the global rebuttal]. Notably, after a few iterations, the weak symmetric model becomes non-symmetric, suggesting that starting close to symmetry is sufficient.
>
> These findings indicate that performance relies more on weight initialization than on exact symmetry. The model's robustness to noise is encouraging, and therefore, the question of biological feasibility focuses on obtaining nearly symmetric initialization. We have ideas for a possible scheme and would be glad to discuss this further if there is interest.
>
>
> > W4
>
> While our model is small compared to Llama and other large models, it is relatively large for biological models, demonstrating results on multi-task learning using the CelebA benchmark with ResNet18.
>
> To further address this issue, we conducted additional experiments comparing our method with the mentioned FA approach [1]. The results show that classic FA scales poorly to ResNet18 on CIFAR. In contrast, our CH learning matches backpropagation performance in the weak symmetric case and learns effectively in the asymmetric case, achieving significantly higher performance than FA. Please refer to the full results attached to the global rebuttal.
>
> > Q1 + Q2
>
> The CH is applied to all layers, including convolutional layers where we use weight sharing, a technique often used in comparing convolutional models with human vision for computational efficiency [2]. This shortcut is supported by prior works showing that position invariance and similar receptive fields emerge in biological, non-convolutional models due to natural image statistics that cause early layers in the visual cortex to be convolution-like [3]. A comparison of deep neural network models and biological networks [4] addresses specifically the use of convolution used by deep neural network models and the actual brain network, expressing the view that the two are similar.
>
> > Q3
>
> We do have the results for standard backpropagation, denoted as "Unit. Scal." in Table 2, following the terminology from the Kurin et al. paper. Thanks for pointing this out; we will refer to it also as "standard backpropagation" in the revised version.
>
> Since we could not compare our results with other biological methods (as none of them uses task selection) we evaluated the results on the benchmark provided by Kurin et al. (2022), which compares various multi-task learning methods on a common architecture and settings. We believe the comparison is somewhat biased in favor of SMTO, as the baseline results include specialized training techniques like early stopping, dropout, and regularization, whereas we use a straightforward 'vanilla' implementation. The results in Table 2 show that our biologically inspired model achieves a similar level of performance, indicating that our model can effectively perform instruction-based learning despite the constraints of a biological model and using the same top-down streams for both learning and guiding attention.
>
> > L1
>
> We will increase the size of the figures in the revised version to better visualize the conducted ablation study results.
>
> The claim "were carefully optimized over the years" refers to advancements in training deep learning models, such as weight initialization schemes and learning rate schedulers, designed specifically for BP and symmetric weights. This raises the question of how optimal these techniques are for the asymmetric case and whether further exploration in asymmetric settings could improve performance.
>
> > L2
>
> Thanks, we will add the reference in the revised version of the paper.
>
> > L3
>
> The legend indeed had an error, we thank you for pointing this out, the word ‘asymmetric’ was inserted by mistake, and will be removed.
>
> > L4
>
> The term is used by recent VLMs, such as Llava [5], that incorporate “visual instruction” training to focus on specific tasks, similar to what we refer to as ‘guided vision’ in our model. For more on the relevance of ‘instructed vision,’ please see the global comment.
>
>
> [1] Lillicrap et al. feedback weights support error backpropagation for deep learning. Nature communications, 7(1): 411 1–10, 2016.
>
> [2] Yamins, Daniel et al. “Performance-optimized hierarchical models predict neural responses in higher visual cortex.” (2014)
>
> [3] Robinson, L., Rolls, E.T. Invariant visual object recognition: biologically plausible approaches. Biol Cybern (2015).
>
> [4] Kriegeskorte, Nikolaus. “Deep neural networks: a new framework for modelling biological vision and brain information processing.” Annu. Rev. Vis. Sci. 2015.
>
> [5] Liu, Haotian, et al. "Visual instruction tuning." NeurIPS (2024).

---

### Author Rebuttal · Authors · 2024-08-06

Many thanks to the reviewers for their useful feedback, the time invested and your effort. Thanks for pointing out weak points, possible improvements, and positive feedback.

Guided vision: we add a general comment about the role of guided vision in our model compared with previous and recent biological and AI models. This is in response to questions asking to elaborate on ‘instructed visual processing’, relation to VLMs, and expanding on the differences with existing biological approaches.

The model is the first to combine two functions in the same top-down stream:  The first is backpropagation, or adjusting synaptic weights, and the second is using top-down (TD) attention. The TD attention guides the visual processing and specifies what to perform (task selection) and where to apply it in the image.

These two TD functions are an essential part of human vision and have been studied in human physiology, anatomy and psychophysics for a long time. However, previous biologically motivated models that used TD for learning, did not include the second important function of TD processing, guiding attention.

Until recently, computer vision models have focused on purely bottom-up inference, i.e. the output of these models was dependent exclusively on the visual input. For instance, models were trained to extract the full scene structure, producing a “scene graph” that represents all the scene components, properties, and relations given an input image [1,2], unlike biological vision, which focuses on selected structures in the scene, guided by TD attention. It is interesting to note that in the recent development of large Vision-Language Models (VLMs), better performance in scene understanding is obtained by using an instructed, or task-guided mode of processing [3,4,5]. In the instructed mode, the model is not required to produce a full image description, but to focus on specific aspects specified by the task.

The comparisons above help to explain and put in context the novelty of the current study and its contribution. For biological vision, the combination of backpropagation-like learning and guided, or instructed visual processing in the TD stream is a major aspect. In current VLMs, instructed visual processing is also proving to be of high value. The current model proposes for the first time an integration of the two tasks in the same TD stream. The combination is relatively simple and natural: the TD network performs error propagation when the TD input provides the error and performs task-selection and guidance when the input is the task or instruction.

[1] D. Xu, Y. Zhu, C. B. Choy, L. Fei-Fei, “Scene graph generation by iterative message passing” in Proceedings–30th IEEE Conference on Computer Vision and Pattern Recognition, CVPR (2017).

[2] G. L. Malcolm, I. I. A. Groen, C. I. Baker, Making sense of real-world scenes. Trends Cogn. Sci. 20, 843–856 (2016).

[3] Liu, Haotian, et al. "Visual instruction tuning." Advances in neural information processing systems 36 (2024).

[4] Dai et al. 2023 “InstructBLIP: Towards General-purpose Vision-Language Models with Instruction Tuning”. In Proceedings of the 37th International Conference on Neural Information Processing Systems (2024).

[5] Shen, Ying, et al. "Multimodal Instruction Tuning with Conditional Mixture of LoRA." arXiv preprint arXiv:2402.15896 (2024).

---

### Decision · Program_Chairs · 2024-09-25

**Decision:**

Accept (poster)

**Comment:**

The paper describe a novel top-down/bottom-up learning rule for visual systems using a counter-Hebbian mechanism. The rule is intended to be a computational model for biological neural networks, allowing attention guided learning via the top-down pathway.

The reviewers appreciated the novelty and the biological plausibility of the proposed work, in particular the fact that it remedies some aspects of classic Hebbian learning that seem much less plausible (backpropagation of the post-synaptic activity). They also liked the extensive validation of the learning rule with several, image-related learning tasks. Some concerns were raised with regard to the similarity to the predictive coding framework, which, however, were well addressed during the rebuttal. One reviewer also criticized the lack of larger scale tests and comparison with some non-biologically motivated rules. Given that the model has not been optimized for performance yet, and is meant to provide a potential explanation for learning in biological systems this criticism seems less severe. Overall, innovative work like this is crucial to push forward our understanding of biological neural information processing.